# REA-RL: Reflection-Aware Online Reinforcement Learning for Efficient Reasoning

**Hexuan Deng**[1,2]   **Wenxiang Jiao**[3*]   **Xuebo Liu**[1*]   **Jun Rao**[1]   **Min Zhang**[1]

[1] Institute of Computing and Intelligence, Harbin Institute of Technology, Shenzhen, China
[2] Zhongguancun Academy, Beijing, China    [3] Xiaohongshu Inc.
`{hxuandeng,wenxiangjiaonju,rao7jun}@gmail.com,`
`{liuxuebo,zhangmin2021}@hit.edu.cn`

## ABSTRACT

Large Reasoning Models (LRMs) demonstrate strong performance in complex tasks but often face the challenge of *overthinking*, leading to substantially high inference costs. Existing approaches synthesize shorter reasoning responses for LRMs to learn, but are inefficient for online usage due to the time-consuming data generation and filtering processes. Meanwhile, online reinforcement learning mainly adopts a length reward to encourage short reasoning responses, but it tends to lose reflection ability and harm performance. To address these issues, we propose REA-RL, which introduces a small reflection model for efficient scaling in online training, offering both parallel sampling and sequential revision. Besides, a reflection reward is designed to further prevent LRMs from favoring short yet non-reflective responses. Experiments show that both methods maintain or enhance performance while significantly improving inference efficiency. Their combination achieves a good balance between performance and efficiency, reducing inference costs by 36% without compromising performance. Further analysis demonstrates that our methods are effective by maintaining reflection frequency for hard problems while appropriately reducing it for easier ones without losing reflection ability. Code is available at `https://github.com/hexuandeng/REA-RL`.

## 1 INTRODUCTION

Large Reasoning Models (LRMs) have demonstrated impressive performance in downstream applications (Jaech et al., 2024; Qwen, 2025; Kilpatrick, 2025). Their human-like deliberation and insightful self-reflection ability facilitate thorough question consideration and verification, thereby improving performance on complex reasoning tasks (Guo et al., 2025; Du et al., 2025). However, this often leads to excessive reasoning with minimal performance benefits, i.e., *overthinking*, substantially increasing the inference cost (Xu et al., 2025; Chen et al., 2025; Qu et al., 2025a).

Prior research (Munkhbat et al., 2025; Xia et al., 2025; Han et al., 2025) attempts to generate shorter reasoning responses for supervised fine-tuning (SFT) or reinforcement learning (RL) to encourage concise response generation. However, this method relies on static datasets, the distribution of which may deviate from the trained model, especially later in training, leading to suboptimal results (Tang et al., 2024). Furthermore, the time-consuming data generation and filtering processes severely limit its feasibility as an online data generation solution.

To address these problems, another line of work (Luo et al., 2025a; Shen et al., 2025; Aggarwal & Welleck, 2025; Liang et al., 2025) employs online reinforcement learning. To enhance the efficiency of online data generation, multiple reasoning paths are sampled in parallel for a single query, with accuracy and length rewards used to encourage correct and concise responses (Zhang & Zuo, 2025; Yeo et al., 2025). However, this self-iterative training paradigm can lead to unpredictable behavior. As illustrated in Figure 1, it can cause the LRM to completely lose its reflection capability, reverting to a naive chain-of-thought style, resulting in suboptimal performance on complex reasoning tasks.

---

* Xuebo Liu and Wenxiang Jiao are corresponding authors.

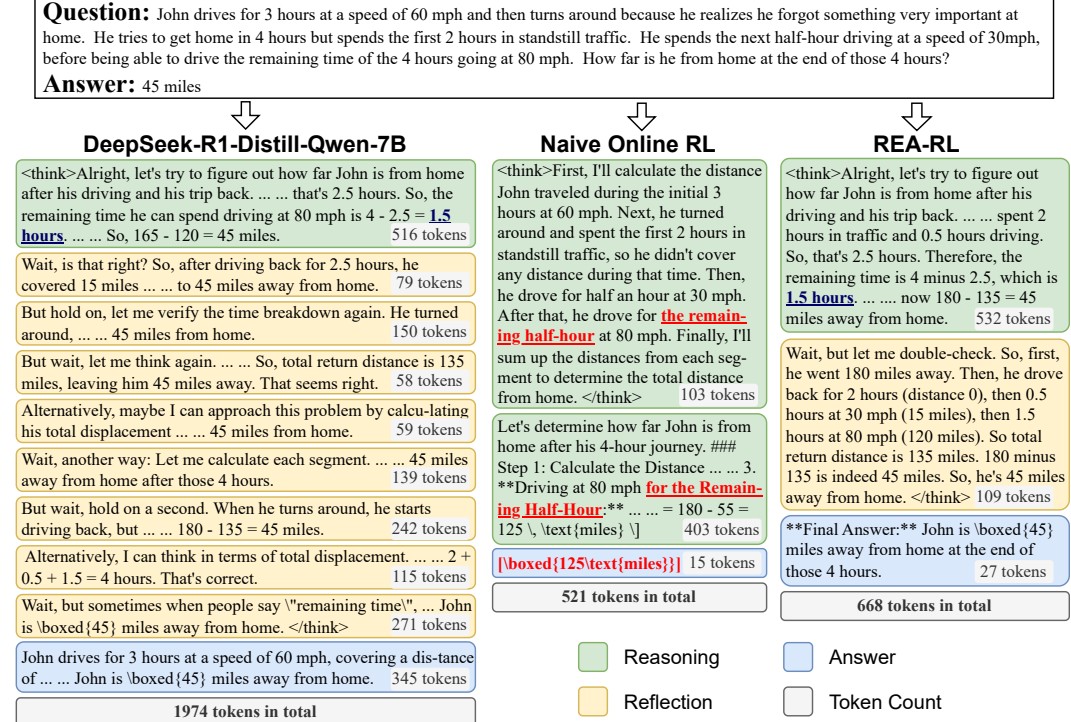

Figure 1: Overthinking and non-reflective cases from GSM8k. The left shows the output of DeepSeek-R1-Distill-Qwen-7B (R1-7B), which reflects eight times before finishing generation. The middle presents the output after online RL training using length rewards, which only spends 103 tokens in "think" part and no reflection, where an error occurs (underlined). The right shows the output of our method, which uses a similar budget to R1-7B in reasoning but only performs a single reflection.

In this paper, we propose REA-RL, a reflection-aware RL framework that integrates the strengths of the aforementioned ones. First, we introduce a small **reflection model** into the online RL process to identify the first reflection position in a sampled response that is very likely to lead to the final answer, and truncate the response to a shorter revision for optimization. It enables us to perform both parallel sampling and sequential revision, which has been demonstrated to achieve computationally optimal test-time scaling during inference (Snell et al., 2024). Second, we introduce a **reflection reward** to prevent the models from favoring short yet non-reflective responses. The reward is calculated based on the density of keywords (e.g., "wait", "but") that signal reflective thinking in the response.

Results reveal that employing only the length reward leads to substantial performance degradation. In contrast, both the reflection model and the reflection reward improve performance and prevent non-reflective behavior. The reflection reward is better at maintaining performance, while the reflection model achieves higher efficiency. Combining the approaches achieves a 36% response shortening across all datasets without performance reduction. Further analysis demonstrates these improvements stem from maintaining the reflection tendency on difficult questions and appropriately reducing reflection on easy questions without losing reflection ability. Our contributions are:

- We design an efficient overthinking detection method, enabling small models to perform this task. Furthermore, we train a **reflection model** for the online generation of shorter response revisions, facilitating both parallel sampling and sequential revision to achieve more efficient scaling.

- We design a **reflection reward** to prevent non-reflective behavior in online RL and significantly enhance model performance compared to using only the length reward.

- Results demonstrate that each method reduces inference cost while preserving model performance. Combining both methods achieves a 36% response shortening without performance degradation.

## 2 RELATED WORK

**Efficient reasoning with response revision.** Training on revised responses with less overthinking via SFT and RL enhances reasoning efficiency. Several methods employ parallel sampling. Munkhbat et al. (2025) construct concise reasoning via best-of-N sampling. Yu et al. (2024) skip reasoning for high-confidence samples, while NoThink (Ma et al., 2025) forces LRMs to skip reasoning for all samples. Other methods utilize stronger models to generate shorter revisions (Kang et al., 2025; Chen et al., 2025; Han et al., 2025; Wen et al., 2025). Still others rely on heuristic algorithms and time-consuming post-verification. SPIRIT-FT (Cui et al., 2025) identifies crucial steps using perplexity. LM-skip (Liu et al., 2024c) stimulates step-skipping behavior by iteratively refining. TokenSkip (Xia et al., 2025) omits less important tokens detected by LLMLingua-2 (Pan et al., 2024). However, their efficiency is often too low for online settings due to complex generation and filtering processes, which incur more than double the generation cost compared to naive sampling.

**Efficient reasoning with length reward.** DeepSeek-R1 (Guo et al., 2025) achieves promising results using RL with verifiable rewards (RLVR). Several methods further introduce length rewards to enhance efficiency, often normalized by a baseline budget. O1-Pruner (Luo et al., 2025a) estimates the budget from a reference model, Liu et al. (2024a) from pairs, while Arora & Zanette (2025) from groups. ShorterBetter (Yi et al., 2025) estimates using shortest correct response, while Arora & Zanette (2025) uses Leave One Out estimator. Kimi K1.5 (Du et al., 2025) normalizes length with the minimum and maximum lengths of generations, while GRPO-LEAD (Zhang & Zuo, 2025) bases it on the length distribution. Other approaches predict the required maximum length budget. DAST (Shen et al., 2025) estimates the budget based on problem difficulty. Yeo et al. (2025) propose a cosine reward that only penalizes excessive length. L1 (Aggarwal & Welleck, 2025) rewards the model for following the length limit in prompts. However, most methods depend solely on multiple samplings of a query, which may cause unpredictable behavior in RL. To address this, we provide online revision and refined reward design, better guiding the model's optimization direction.

## 3 OVERTHINKING DETECTION

Previous works (Du et al., 2025; Chen et al., 2025; Xu et al., 2025; Qu et al., 2025a) observe that LRMs like QwQ-32B-Preview (Qwen, 2025) and DeepSeek-R1 (Guo et al., 2025) tend to generate more solution rounds for easy math problems, allocating excessive computational resources with limited utility. We extend this analysis to smaller LRMs, and propose an effective detection method that does not necessitate the use of powerful closed-source LLMs for detection.

### 3.1 AUTO-DETECTION OF OVERTHINKING

**Problem definition.** As illustrated in Figure 1, we observe a similar phenomenon in R1-7B: the model tends to engage in excessive reflection after completing reasoning and obtaining the correct answer, leading to overthinking. In preliminary experiments, we find that weaker LLMs struggle to extract the boundaries of each reflection. However, we note that the conclusion of both reasoning and reflection is often a restatement of the answer. Therefore, we prompt LLMs to determine if each part of the response contains the correct answer, which is easier for LLMs. The thought process preceding the first occurrence of the correct answer is considered effective reasoning, while each subsequent appearance is an additional reflection, i.e., overthinking.

**Detection method.** Given inconsistent formatting and incidental early mentions of the answer in reasoning, regex-based extraction is unreliable. Thus, we employ Qwen2.5-32B-Instruct (Qwen-32B; Yang et al., 2024a) to identify these positions within the "think" part of LRMs. Specifically, we segment the think part into smaller chunks and provide the question along with these chunks, prompting the model to determine whether each chunk contains the correct answer. To ensure accuracy, we apply a filtering step: chunks containing the answer are verified again one by one with Qwen-32B, and only those confirmed as positive in both detections are considered correct. We design prompts with and without the inclusion of the gold answer as input, detailed in Appendix A.

**Response revision.** The tokens after the first correct answer are identified as overthinking and subsequently removed. Following this, we generate the revision by forcibly terminating the "think"

| Method | GSM8K | | Math500 | | Gaokao23 | | Amc23 | | Aime24 | | Average | |
|---|---|---|---|---|---|---|---|---|---|---|---|---|
| | $Acc\uparrow$ | $TR\downarrow$ | $Acc\uparrow$ | $TR\downarrow$ | $Acc\uparrow$ | $TR\downarrow$ | $Acc\uparrow$ | $TR\downarrow$ | $Acc\uparrow$ | $TR\downarrow$ | $Acc\uparrow$ | $TR\downarrow$ |
| Original | 91.66 | $100_{1344}$ | 92.00 | $100_{3893}$ | 81.82 | $100_{3785}$ | 88.12 | $100_{5840}$ | 48.33 | $100_{10460}$ | 80.39 | $100_{5064}$ |
| Fixed Trunc (90) | 89.39 | 80.51 | 91.40 | 83.51 | 81.30 | 83.20 | 85.00 | 85.68 | 44.17 | 88.72 | 78.25 | 84.32 |
| Fixed Trunc (80) | 88.25 | 72.47 | 89.60 | 75.52 | 79.22 | 75.22 | 78.44 | 77.67 | 37.08 | 80.09 | 74.52 | 76.19 |
| Fixed Trunc (70) | 87.26 | 64.73 | 86.00 | 66.76 | 75.58 | 66.61 | 73.12 | 68.99 | 33.33 | 71.02 | 71.06 | 67.62 |
| Model Revise | 89.46 | 62.43 | 93.20 | 71.85 | 80.52 | 70.41 | 89.06 | 79.95 | 50.83 | 93.59 | **80.61** | **75.65** |
| + Gold | 88.78 | 54.09 | 92.00 | 57.80 | 79.48 | 59.71 | 87.19 | 67.71 | 51.67 | 92.31 | 79.82 | 66.32 |

Table 1: Performance and efficiency after model revision. "Original" denotes the R1-7B performance before revision. "Acc" represents the response accuracy, and "TR" represents the efficiency, calculated as its token cost divided by that of "Original". Therefore, the TR in the first row is 100, and its subscript indicates its token cost. "Average" is the macro-average across datasets. The best and second-best results in the "Average" column are marked **bold** and underline, respectively. Considering the trade-off between TR and Acc, for TR, only methods whose accuracy does not decrease are **bolded** or underlined. Abbreviations of the methods are defined in §3.2.

part and compelling the LRM to continue generating the final answer, prefixed with "**Final Answer:**", which yields a revision without overthinking. For verification, the revision is deemed correct if the LRM can accurately generate the final answer. Finally, to prevent excessively long generations from skewing the average length, we limit the generation budget to 16k tokens.

## 3.2 EFFICIENCY OF AUTO-DETECTION

**Experimental setup.** We first use R1-7B to answer each question, and then apply the above response revision method from the previous section to truncate the generated reasoning path and complete it with R1-7B. We evaluate the correctness of response revision by the accuracy of the final answers after completion. "Model Revise" and "+Gold" represent revision using our method without and with the gold answer as input, respectively. Our baseline for comparison is "Fixed Trunc $(n)$", which denotes fixed truncation of the original generated response, retaining $n\%$ of the thinking tokens and truncating the response at the nearest newline. Subsequently, similar to response revision, the "think" part is terminated, and the LRM is compelled to generate the final answer. Results before and after revision are in Table 1. The evaluated math datasets are introduced in Appendix B.2, and are ordered from left to right with increasing difficulty.

**Awesome auto-detection ability.** Without the gold answer as input, we can automatically remove 24% of tokens without performance degradation. When provided, 34% of tokens are removed with a minor performance decrease. However, fixed truncation shows a considerable performance decline with increasing truncation ratios. These results demonstrate the effectiveness of our detection method and underscore the severity of the overthinking problem.

**Greater overthinking on easier problems.** Across three easier and two harder datasets, overthinking tokens our method detects decrease progressively, at 37% and 17%, respectively. This indicates that overthinking is more prevalent in easier datasets and less prevalent in more challenging ones.

## 4 REFLECTION-AWARE ONLINE REINFORCEMENT LEARNING

To maintain model performance while reducing inference costs, a fundamental method is to employ **online RL** with a **length reward** (§4.1), which ensures distributional alignment between the data and the model. Building upon this, we make improvements from two perspectives. First, based on the detection method in §3, we introduce a **reflection model** that provides revisions online (§4.2). This not only augments the data but also provides shorter, non-overthinking paths that serve as positive examples for training, which are lacking in parallel sampling. Second, we refine the reward by including a **reflection reward** to penalize non-reflective behavior (§4.3), which would otherwise lead to a significant performance drop. The workflow is illustrated in Figure 2.

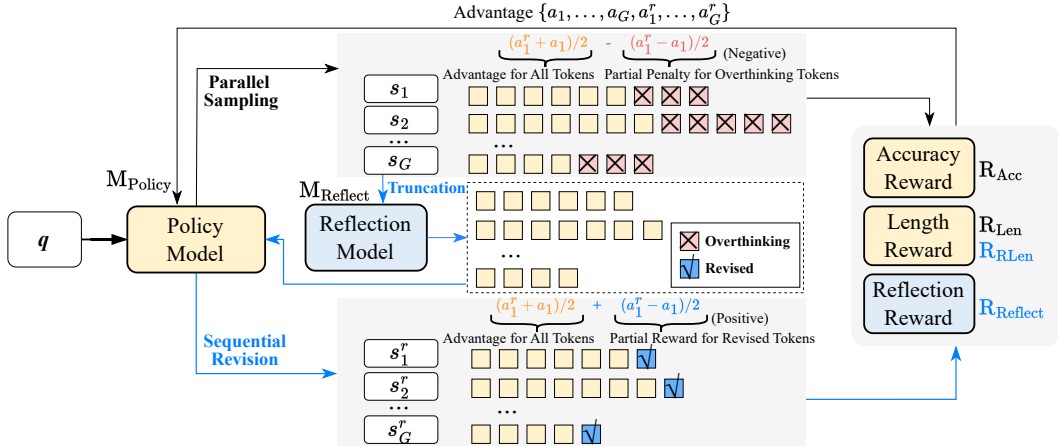

Figure 2: Workflow of REA-RL. We first *parallel sample* $G$ paths as in GRPO. Then, the reflection model identifies and truncates overthinking tokens (red ones with ×), retaining the preceding yellow segments. After that, the policy model finishes the truncated paths, generating revised tokens (blue ones with ✓), and yielding $G$ revised paths. Finally, both the $G$ original and $G$ revised paths are used for training. Cases are in Appendix B.1. In addition to the naive accuracy reward $R_{Acc}$ and length reward $R_{Len}$, we refine the length reward ($R_{RLen}$) and introduce a new reflection reward ($R_{Reflect}$).

## 4.1 ONLINE REINFORCEMENT LEARNING

**Online RL.** We adopt Grouped Relative Policy Optimization (GRPO, Shao et al., 2024). Specifically, for each question in a given dataset, GRPO samples a group of paths $S = \{s_1, ..., s_G\}$ in parallel from the policy model. Then, for each path $s_i$, we calculate its reward $r_i$ as the sum of several reward functions introduced later. These rewards are normalized within the group $S$ to get the advantage of each path, i.e., $a_i = \frac{r_i - \text{mean}(\mathbf{r})}{\text{std}(\mathbf{r})}$. Finally, the policy model is optimized by increasing the probability of paths with high advantage and decreasing the probability of those with low advantage.

Following Shao et al. (2024), we apply the rule-based accuracy reward $R_{Acc}$ to mitigate reward hacking. Specifically, we extract the final answer and compare it with the gold answer to verify its correctness. The reward is 1 for a correct answer and 0 otherwise.

**Length reward.** Following Kimi K1.5 (Du et al., 2025), we incorporate a length reward to improve efficiency. Formally, given a group of sampled responses $\{s_1, ..., s_G\}$, where max_len is the length of the longest response and min_len is that of the shortest, the length reward for the $i$-th response is:

$$R_{Len}(s_i) = \begin{cases} \lambda & \text{if } s_i \text{ is correct} \\ \min(0.5, \lambda) & \text{if } s_i \text{ is incorrect} \end{cases}, \text{ where } \lambda = 1 - \frac{\text{len}(s_i) - \text{min\_len}}{\text{max\_len} - \text{min\_len}}. \quad (1)$$

## 4.2 REFLECTION MODEL FOR ONLINE SEQUENTIAL REVISION

**Design principle.** We use a reflection model to produce shorter online revisions of the trained model's reasoning path, using the same revision definition as §3.1. The task is simple, but enforcing per-sentence judgments over long traces is challenging. As evaluated in §3.2, a 32B LLM performs well with a two-step annotation, but is too slow. Qwen2.5-7B-Instruct (Qwen-7B) is faster, yet in 17% of cases its outputs violate the required judgment count. Given the task's simplicity, we distill the 32B two-step revision ability into Qwen-7B with SFT and run it in one step to ensure efficiency.

**Reflection model training.** To construct SFT data, we generate four responses per question on the training dataset using R1-7B, retaining only correct responses. Each chunk within these responses is labeled based on whether it contains the correct answer following §3.1. We then construct the SFT data using the question and all response chunks as input, training the reflection model to predict the category of each chunk in a single step. For efficiency, we adopt a concise output format, directly generating the ID and its category without additional deliberation, as detailed in Appendix A.

**Online RL with sequential revision.** As depicted in Figure 2, following the parallel sampling of multiple responses in online RL, we utilize the reflection model for sequential revision. Specifically, for the sampled responses $S = \{s_1, ..., s_G\}$, we first use the reflection model $M_{\text{Reflect}}$ to remove overthinking tokens (red segment), copy the preceding segments from the original responses $S$ (yellow segment), and prompt the policy model $M_{\text{Policy}}$ to generate a final answer (blue segment), which forms the revision $S^r = \{s_1^r, ..., s_G^r\}$. Cases are shown in Appendix B.1. Ultimately, both the original responses $S$ and the revised responses $S^r$ are used as training data for the policy model, enabling an additional dimension of scaling and guiding the optimization direction of RL. Formally:

$$\{s_1, ..., s_G, s_1^r, ..., s_G^r\} \xrightarrow{\text{Online RL}} M_{\text{Policy}}, \text{ where } s_i^r = M_{\text{Policy}}(M_{\text{Reflect}}(s_i)). \tag{2}$$

**Equivalence to a partial advantage.** We find that our method is equivalent to a partial penalty only for the overthinking portion. Given the substantial similarity between the pre- and post-revision responses, we can decompose the advantage into two components. Let $a_i$ and $a_i^r$ represent the advantages of the original response $s_i$ and revised response $s_i^r$, respectively. The first component is the average advantage for all tokens in both responses, formulated as $(a_i^r + a_i)/2$. The second component is a partial advantage for the different parts. The overthinking tokens (red) receive a penalty of $-(a_i^r - a_i)/2$, while the revised tokens (blue) receive $(a_i^r - a_i)/2$. When revision is successful, the length reward ensures $a_i^r > a_i$, effectively penalizing overthinking.

However, if revision transforms a correct answer into an incorrect one, the opposite effect occurs, potentially encouraging overthinking. Furthermore, if both the original and revised responses are incorrect, it suggests a need for more extensive reasoning, making a preference for shorter outputs detrimental. Therefore, we discard the revised responses in such cases and retain the original ones.

### 4.3 REFLECTION-AWARE REWARD REFINEMENT

**Reflection reward.** To prevent the models from favoring short yet non-reflective responses due to the length reward, we introduce a reflection reward based on the presence of reflective tokens, i.e., "wait", "alternatively", "check", and "but". We calculate the density of these tokens, and count closely clustered occurrences as a single instance to prevent consecutive reflective tokens within a single reflection. Formally, for the current response $s_i$, let $N_{\text{Token}}$ be the total number of tokens in the response and $N_{\text{Reflect}}$ be the number of reflective tokens. The reflection reward $R_{\text{Reflect}}$ is:

$$R_{\text{Reflect}}(s_i) = \min(0, \frac{D_i}{D_{0.2}} - 1), \text{ where } D_i = \frac{N_{\text{Reflect}}}{N_{\text{Token}}}. \tag{3}$$

Here, $D_{0.2}$ represents the 0.2 quantile of the reflection density in the training data, and $D_i$ is the density of $s_i$. This reward penalizes responses only when their density falls within the lowest 20% of observed densities, thereby preventing the reward from inadvertently promoting overthinking. Empirically, using any quantile $\leq 0.2$ yields similarly strong performance; a smaller quantile places greater emphasis on efficiency, whereas a larger one trades efficiency for accuracy.

**Length reward refinement.** The length reward in Kimi K1.5 demonstrates a bias towards shorter responses even when incorrect. However, for challenging queries, more extensive reasoning is often required, conflicting with the current reward mechanism. Thus, we follow Zhang & Zuo (2025) to set the length reward to zero if the response is incorrect, i.e.:

$$R_{\text{RLen}}(s_i) = \begin{cases} \lambda & \text{if } s_i \text{ is correct} \\ 0 & \text{if } s_i \text{ is incorrect} \end{cases}, \text{ where } \lambda = 1 - \frac{\text{len}(s_i) - \text{min\_len}}{\text{max\_len} - \text{min\_len}}. \tag{4}$$

## 5 EXPERIMENTS

### 5.1 EXPERIMENTAL SETUP

We maintain most settings across experiments, with training configs in Appendix B.2. Baselines are:

**Online RL training.** "$M_{\text{Reflect}}$" represents online revision in §4.2. "$R_{\text{Len}}$" represents adding the Kimi K1.5 length reward, "$R_{\text{RLen}}$" represents adding our refined length reward, and "$R_{\text{Reflect}}$" represents adding the reflection reward in §4.3. Accuracy reward is used in all GRPO settings. The primary baselines are 1) **GRPO**, which uses accuracy reward only. 2) **GRPO $R_{\text{Len}}$**, which uses accuracy reward along with the Kimi K1.5 length reward.

| Method | GSM8K | | Math500 | | Gaokao23 | | Amc23 | | Aime24 | | Average | |
|---|---|---|---|---|---|---|---|---|---|---|---|---|
| | Acc↑ | TR↓ | Acc↑ | TR↓ | Acc↑ | TR↓ | Acc↑ | TR↓ | Acc↑ | TR↓ | Acc↑ | TR↓ |
| **Original** | 91.58 | $100_{1312}$ | 88.40 | $100_{3389}$ | 77.40 | $100_{3270}$ | 77.50 | $100_{4746}$ | 40.83 | $100_{6815}$ | 75.14 | $100_{3906}$ |
| **SFT** | 89.92 | 64.63 | 89.20 | 69.25 | 77.92 | 72.91 | 82.81 | 72.67 | 41.25 | 89.98 | 76.22 | 73.89 |
| **RPO** | 90.14 | 51.22 | 89.80 | 53.35 | 83.12 | 55.20 | 80.62 | 59.54 | 41.67 | 74.78 | 77.07 | 58.82 |
| **GRPO** | 92.80 | 117.99 | 89.00 | 100.74 | 79.22 | 101.31 | 79.69 | 96.23 | 39.17 | 98.88 | 75.98 | 103.03 |
| **GRPO $R_{Len}$ (Du et al., 2025)** | 85.97 | 32.01 | 87.60 | 59.04 | 70.91 | 58.84 | 85.62 | 71.43 | 45.42 | 87.42 | 75.10 | 61.75 |
| NoThink (Ma et al., 2025) | 85.06 | 20.50 | 83.20 | 27.94 | 65.71 | 25.81 | 65.00 | 28.82 | 20.42 | 44.37 | 63.88 | 29.49 |
| ShorterBetter (Yi et al., 2025) | 85.37 | 15.17 | 83.40 | 29.30 | 66.75 | 25.96 | 76.56 | 43.43 | 48.33 | 68.67 | 72.08 | 36.51 |
| Light-R1 (Wen et al., 2025) | 86.28 | 39.25 | 87.00 | 77.40 | 68.05 | 80.89 | 75.31 | 97.01 | 42.92 | 96.92 | 71.91 | 78.29 |
| DAST (Shen et al., 2025) | 87.79 | 34.60 | 89.20 | 77.49 | 81.30 | 78.62 | 84.06 | 87.40 | 42.50 | 96.71 | 76.97 | 74.96 |
| Arora & Zanette (2025) | 90.45 | 40.32 | 91.20 | 73.21 | 76.10 | 70.55 | 82.50 | 84.68 | 43.33 | 94.34 | 76.72 | 72.62 |
| **GRPO $R_{Len}$ $M_{Reflect}$** | 89.23 | 37.80 | 91.20 | 48.66 | 78.44 | 52.23 | 86.56 | 59.65 | 41.67 | 84.15 | 77.42 | **56.50** |
| **GRPO $R_{RLen+Reflect}$** | 92.72 | 67.07 | 91.40 | 69.52 | 80.52 | 72.54 | 85.62 | 75.64 | 47.92 | 91.37 | **79.64** | 75.23 |
| **GRPO $R_{RLen+Reflect}$ $M_{Reflect}$** | 89.99 | 53.12 | 89.60 | 61.79 | 81.30 | 61.19 | 85.62 | 71.39 | 42.92 | 88.45 | 77.89 | 67.19 |
| **Original** | 91.66 | $100_{1344}$ | 92.00 | $100_{3893}$ | 81.82 | $100_{3785}$ | 88.12 | $100_{5840}$ | 48.33 | $100_{10460}$ | 80.39 | $100_{5064}$ |
| **SFT** | 89.99 | 68.75 | 90.60 | 69.25 | 79.48 | 74.37 | 88.44 | 72.11 | 48.75 | 89.68 | 79.45 | 74.83 |
| **RPO** | 90.14 | 50.00 | 89.80 | 46.90 | 83.12 | 47.77 | 82.19 | 50.75 | 42.92 | 51.41 | 77.63 | 49.37 |
| **GRPO** | 92.87 | 116.52 | 93.40 | 99.20 | 82.34 | 100.26 | 87.19 | 97.31 | 50.42 | 97.18 | 81.24 | 102.09 |
| **GRPO $R_{Len}$ (Du et al., 2025)** | 85.97 | 31.25 | 88.40 | 56.43 | 72.21 | 55.88 | 87.81 | 65.65 | 50.00 | 76.96 | 76.88 | 57.23 |
| NoThink (Ma et al., 2025) | 85.06 | 20.01 | 84.20 | 27.23 | 66.23 | 24.70 | 66.25 | 26.10 | 23.33 | 38.38 | 65.01 | 27.28 |
| ShorterBetter (Yi et al., 2025) | 85.37 | 14.81 | 84.20 | 27.02 | 66.75 | 22.85 | 76.88 | 37.52 | 50.00 | 51.59 | 72.48 | 30.76 |
| Light-R1 (Wen et al., 2025) | 86.28 | 38.32 | 91.60 | 77.40 | 74.55 | 81.88 | 88.75 | 96.73 | 54.17 | 88.15 | 79.07 | 76.50 |
| DAST (Shen et al., 2025) | 87.79 | 34.67 | 91.40 | 79.84 | 83.12 | 80.79 | 88.44 | 87.12 | 51.67 | 98.41 | 80.48 | 76.17 |
| Arora & Zanette (2025) | 90.45 | 39.58 | 93.20 | 70.15 | 77.14 | 70.20 | 86.88 | 83.39 | 48.33 | 90.25 | 79.20 | 70.71 |
| **GRPO $R_{Len}$ $M_{Reflect}$** | 89.23 | 36.90 | 92.40 | 45.11 | 79.74 | 47.50 | 88.12 | 56.04 | 47.92 | 75.82 | 79.48 | 52.27 |
| **GRPO $R_{RLen+Reflect}$** | 92.72 | 65.48 | 92.80 | 66.71 | 81.82 | 68.27 | 88.75 | 72.17 | 54.58 | 86.21 | **82.13** | 71.77 |
| **GRPO $R_{RLen+Reflect}$ $M_{Reflect}$** | 89.99 | 52.08 | 91.00 | 60.78 | 82.08 | 55.85 | 89.38 | 67.76 | 51.25 | 81.09 | 80.74 | **63.51** |

*(Left margin labels: "Budget: 8k" for the first block, "Budget: 16k" for the second block.)*

Table 2: Main results of our proposed methods. Most abbreviations align with Table 1. Baseline definitions are in §5.1. "GRPO $R_{Len}$ $M_{Reflect}$" represents the addition of the reflection model, "GRPO $R_{RLen+Reflect}$" represents the addition of the reward optimization, and "GRPO $R_{RLen+Reflect}$ $M_{Reflect}$" represents the combination of both optimizations. "Budget" is the max tokens allowed per question.

**Offline training.** We also compare with commonly used offline baselines. We use the revised responses generated from the 32B model as the training data, following §3, with other generation settings consistent with the online approach. Only data that the answer is correct after revision is used for training. We then conduct training using 1) **SFT** (Zhang et al., 2023): We simply fine-tune the model using the aforementioned revised dataset. 2) **RPO** (Pang et al., 2024): We treat the revised responses as positive examples and the original responses as negative examples.

For further comparison, we evaluate against related methods initialized from R1-7B, including DAST (Shen et al., 2025), Light-R1 (Wen et al., 2025), ShorterBetter (Yi et al., 2025), and Arora & Zanette (2025). We download their publicly available checkpoints and run the same evaluation. For the prompt-based method NoThink (Ma et al., 2025), we use its released prompt configuration.

**Evaluation dataset.** We follow Yang et al. (2024b) in evaluating our approach on five math test sets, ordered by difficulty: **GSM8K** (Cobbe et al., 2021), 8.5K grade school math problems; **MATH500** (Hendrycks et al., 2021b), 500 challenging high school competition problems; **Gaokao23** (Liao et al., 2024), English-translated math questions from the 2023 Chinese Gaokao exam; **Amc23**[1], 2023 American Mathematics Competitions; and **Aime24**[2], 2024 American Invitational Mathematics Examination. We employ the math validator provided by rStar (Guan et al., 2025). Considering the limited size of Amc23 and Aime24, we sample 8 paths for each question to mitigate randomness. We report additional experimental results on more models and evaluation domains in Appendix B.3.

**Reflection model training.** We employ R1-7B to generate reasoning processes on DeepScaleR, sampling four paths for each question. In 74.03% of the questions, the model generates at least one correct answer in 8k tokens, and in 46.54% of the questions, all four answers are correct. Subsequently, we label the paths with correct answers as in §3.1 to form the training data. We then fine-tune Qwen2.5-7B-Instruct for one epoch. During online sequential revision, we set the temperature to 0.

---

[1] https://huggingface.co/datasets/AI-MO/aimo-validation-amc
[2] https://huggingface.co/datasets/AI-MO/aimo-validation-aime

## 5.2 Main Result

**Improved performance compared to offline method.** The results are in Table 2. The final three rows showcase the performance of the reflection model, reward refinement, and their combined approach. While RPO shares similar ideas with the sequential revision, the primary distinction lies in the offline dataset. RPO achieves comparable performance at an 8k budget but significantly degrades on two more challenging datasets at a 16k budget. In contrast, our method demonstrates better performance, highlighting the advantage of online training over offline training.

**Improved efficiency and performance balance compared to online methods.** GRPO, using only an accuracy reward, shows no significant performance improvement, suggesting that gains are not solely due to more training. Adding a length reward greatly reduces inference costs but severely hurts performance. Relative to all other baselines including GRPO $R_{Len}$, we consistently obtain higher accuracy at comparable truncation ratios. While some baselines achieve notably more aggressive truncation, they incur large accuracy drops that undermine their practicality.

**Reflection model and reflection reward target distinct dimensions.** The reflection model is more effective in reducing response length, whereas reflection reward contributes more to accuracy enhancement. By integrating both strategies, a balanced outcome can be achieved, yielding a 36% efficiency improvement without compromising performance. However, the improvement under an 8k budget remains limited. This can be attributed to the advantage of shorter generations under a constrained budget, and the reflection model's inherent ability to foster reflection, as analyzed in §5.4. The effectiveness of $R_{RLen}$ and $R_{Reflect}$ and the choice of quantile is verified in Appendix B.4.

## 5.3 Reflection Model Analysis

**Experimental setup.** We follow §3.2 to verify our reflection model $M_{Reflect}$. We use $M_{Reflect}$ to truncate paths generated by R1-7B, then have R1-7B produce the final answer. The truncation is validated by the correctness of this answer. We compare against the untrained 7B and 32B LLMs and a fixed truncation strategy. For $M_{Reflect}$, we define three revision strengths: **Normal** truncates at the first identified correct answer; **Weak** truncates at the second such position; and **Strong** truncates before the first position where the truncation probability exceeds 0.25. If no such position exists, the Normal position is used. **Fixed Trunc** truncates at the closest sentence-ending position matching the truncation ratio achieved by our method, and then prompts R1-7B to generate the final answer.

**Comparable performance to 32B at a significantly lower cost.** Results are shown in Table 3. Our reflection model does not incorporate the gold answer as input. It significantly outperforms its 7B base model (7B Revise). In comparison to the 32B model revised without gold input, our Strong strategy achieves a comparable compression ratio while exhibiting minimal performance degradation, demonstrating that our model achieves comparable results at a substantially reduced cost. Furthermore, our method also outperforms fixed truncation, especially when the truncation ratio is large. Finally, we observe that under certain evaluations, the accuracy actually increases after revision. This indicates that the model may have already arrived at the correct answer, but fails to explicitly state this conclusion under the 16k token budget.

**REA-RL outperforms inference with reflection model.** To ensure revision accuracy, we employ the Normal strategy during training. However, REA-RL with the reflection model achieves a higher compression ratio than the Strong strategy while maintaining comparable performance, despite the increased inference cost associated with the generate-then-revise approach of the Strong strategy. This improvement can be attributed to the length reward and the online training, which iteratively train the model and provide further guidance for shortening already concise responses.

**Online revision as an efficient scaling strategy.** The reflection model provides revised responses, which doubles the dataset size. To evaluate whether other methods could yield similar benefits, we extend the GRPO baseline generation to 8 paths, i.e., "Gen8", to verify whether scaling parallel sampling is more effective. However, Gen8 indicates no consistent improvement over parallel sampling of 4 paths, while our combination with sequential revision, under the same reward $R_{Len}$,

| Method | GSM8K | | Math500 | | Gaokao23 | | Amc23 | | Aime24 | | Average | |
|---|---|---|---|---|---|---|---|---|---|---|---|---|
| | Acc↑ | TR↓ | Acc↑ | TR↓ | Acc↑ | TR↓ | Acc↑ | TR↓ | Acc↑ | TR↓ | Acc↑ | TR↓ |
| Original | 91.66 | $100_{1344}$ | 92.00 | $100_{3893}$ | 81.82 | $100_{3785}$ | 88.12 | $100_{5840}$ | 48.33 | $100_{10460}$ | 80.39 | $100_{5064}$ |
| **7B Revise** | 86.50 | 63.32 | 89.20 | 64.60 | 77.40 | 66.16 | 82.81 | 73.17 | 42.92 | 80.98 | 75.77 | 69.65 |
| + Gold | 85.90 | 55.36 | 85.60 | 56.59 | 76.62 | 55.96 | 79.06 | 61.25 | 42.92 | 76.58 | 74.02 | 61.15 |
| **32B Revise** | 89.46 | 62.43 | 93.20 | 71.85 | 80.52 | 70.41 | 89.06 | 79.95 | 50.83 | 93.59 | 80.61 | 75.65 |
| + Gold | 88.78 | 54.09 | 92.00 | 57.80 | 79.48 | 59.71 | 87.19 | 67.71 | 51.67 | 92.31 | 79.82 | 66.32 |
| $M_{Reflect}$ **Weak** | 90.75 | 84.82 | 92.40 | 86.64 | 82.86 | 86.71 | 88.75 | 88.90 | 50.00 | 93.10 | 80.95 | 88.03 |
| **Fixed Trunc (88.47)** | 89.84 | 84.23 | 92.00 | 86.75 | 81.04 | 86.92 | 88.12 | 89.30 | 47.50 | 93.55 | 79.70 | 88.15 |
| $M_{Reflect}$ **Normal** | 90.22 | 79.02 | 91.80 | 81.09 | 82.86 | 81.82 | 89.06 | 83.80 | 49.17 | 91.75 | 80.62 | 83.50 |
| **Fixed Trunc (84.17)** | 89.23 | 78.50 | 91.40 | 82.02 | 80.52 | 82.51 | 86.25 | 85.14 | 46.67 | 92.48 | 78.81 | 84.13 |
| $M_{Reflect}$ **Strong** | 90.07 | 72.99 | 92.00 | 75.65 | 82.08 | 77.75 | 87.81 | 77.16 | 48.75 | 89.04 | 80.14 | 78.52 |
| **Fixed Trunc (78.93)** | 88.32 | 72.77 | 90.00 | 76.73 | 79.22 | 78.76 | 80.94 | 80.02 | 45.83 | 90.12 | 76.86 | 79.68 |
| **GRPO** | 92.87 | 116.52 | 93.40 | 99.20 | 82.34 | 100.26 | 87.19 | 97.31 | 50.42 | 97.18 | 81.24 | 102.09 |
| **GRPO Gen8** | 92.42 | 103.35 | 92.00 | 97.82 | 82.34 | 100.24 | 90.00 | 92.74 | 50.00 | 93.60 | 81.35 | 97.55 |
| **GRPO $R_{Len}$** | 85.97 | 31.25 | 88.40 | 56.43 | 72.21 | 55.88 | 87.81 | 65.65 | 50.00 | 76.96 | 76.88 | 57.23 |
| **GRPO $R_{Len}$ Gen8** | 85.52 | 25.74 | 88.20 | 51.68 | 72.21 | 56.78 | 83.12 | 63.78 | 53.33 | 82.51 | 76.48 | 56.10 |
| **GRPO $R_{Len}$ $M_{Reflect}$** | 89.23 | 36.90 | 92.40 | 45.11 | 79.74 | 47.50 | 88.12 | 56.04 | 47.92 | 75.82 | 79.48 | 52.27 |
| **GRPO $R_{RLen+Reflect}$** | 92.72 | 65.48 | 92.80 | 66.71 | 81.82 | 68.27 | 88.75 | 72.17 | 54.58 | 86.21 | **82.13** | 71.77 |
| **GRPO $R_{RLen+Reflect}$ $M_{Reflect}$** | 89.99 | 52.08 | 91.00 | 60.78 | 82.08 | 55.85 | 89.38 | 67.76 | 51.25 | 81.09 | 80.74 | **63.51** |

(Budget: 16k)

Table 3: Results of our proposed reflection model. "7B Revise" and "32B Revise" refer to the two-step revision method introduced in §3, using Qwen-7B and Qwen-32B without training. "$M_{Reflect}$" uses our 7B reflection model with one-step revision. "Fixed Trunc" denotes a fixed truncation strategy with the same ratio as our method. Truncation strengths are defined in §5.3.

| Method | GSM8K | | Math500 | | Gaokao23 | | Amc23 | | Aime24 | | Average | |
|---|---|---|---|---|---|---|---|---|---|---|---|---|
| | Acc↑ | Reflect↓ | Acc↑ | Reflect↓ | Acc↑ | Reflect↓ | Acc↑ | Reflect↓ | Acc↑ | Reflect↓ | Acc↑ | Reflect↓ |
| Original | 91.66 | 105.48 | 92.00 | 107.75 | 81.82 | 90.94 | 88.12 | 102.72 | 48.33 | 84.82 | 80.39 | 98.34 |
| **SFT** | 89.99 | 87.73 | 90.60 | 104.43 | 79.48 | 79.97 | 88.44 | 90.00 | 48.75 | 82.30 | 79.45 | 88.89 |
| **RPO** | 90.14 | 156.68 | 89.80 | 146.65 | 83.12 | 130.13 | 82.19 | 127.03 | 42.92 | 100.61 | 77.63 | 132.22 |
| **GRPO** | 92.87 | 97.50 | 93.40 | 107.52 | 82.34 | 88.69 | 87.19 | 98.06 | 50.42 | 82.28 | 81.24 | 94.81 |
| **GRPO $R_{Len}$** | 85.97 | 813.96 | 88.40 | 128.32 | 72.21 | 114.74 | 87.81 | 120.09 | 50.00 | 99.07 | 76.88 | 255.24 |
| **GRPO $R_{Len}$ $M_{Reflect}$** | 89.23 | 194.97 | 92.40 | 135.54 | 79.74 | 120.05 | 88.12 | 107.91 | 47.92 | 89.34 | 79.48 | 129.56 |
| **GRPO $R_{RLen+Reflect}$** | 92.72 | 141.63 | 92.80 | 118.61 | 81.82 | 99.24 | 88.75 | 106.08 | 54.58 | 86.13 | **82.13** | 110.34 |
| **GRPO $R_{RLen+Reflect}$ $M_{Reflect}$** | 89.99 | 150.87 | 91.00 | 126.51 | 82.08 | 109.91 | 89.38 | 105.89 | 51.25 | 90.25 | 80.74 | 116.69 |

(Budget: 16k)

Table 4: Reflection density of REA-RL and baselines. "Reflect" represents the average number of tokens between each reflective token, i.e., a smaller value indicates more frequent reflection.

demonstrates superior performance and truncation ratios, thereby establishing the enhanced efficacy of our scaling method compared to mere parallel sampling.

Regarding training time, we optimize the implementation by deferring the update of the vllm inference model, which enables parallel execution of training and data generation, resulting in approximately a twofold speedup. On 3 NVIDIA A800 80G GPUs, GRPO with sampling 4 paths requires 80 hours, while sampling 8 paths requires 110 hours. Our method, incorporating the reflection model, requires 120 hours. Given the significant performance improvement, this additional computational cost is deemed acceptable. Furthermore, our approach actually reduces token usage and can lower cost in large-scale experiments, detailed in Appendix B.5. Finally, if training efficiency is a primary concern, utilizing the reflection reward alone also requires only 80 hours and achieves improved performance.

## 5.4 REFLECTION ABILITY ANALYSIS FOR THE TRAINED MODEL

**Length reward impacts reflection frequency.** We calculate the average number of tokens between reflective tokens in the responses, and the results are in Table 4. Introducing only a length reward in online RL leads to a severe decrease in the frequency of reflective tokens on easy problems. However, for challenging problems, the model retains its reflection capability, as reflection is crucial for obtaining the accuracy reward. An example is shown in the middle of Figure 1, where the solution only performs planning in the "think" part and solves the problem without any reflection. Due to the minimal token consumption, an error occurs during planning. In contrast, our method preserves the style of R1 by reasoning with sufficient tokens, and performing reflection after reasoning.

**Both reflection model and reflection reward enhance reflection.** Across the three easier datasets, our three approaches yield an average 47% increase in reflection probability and a 5% performance improvement over GRPO $R_{Len}$, demonstrating that REA-RL mitigates non-reflective behavior.

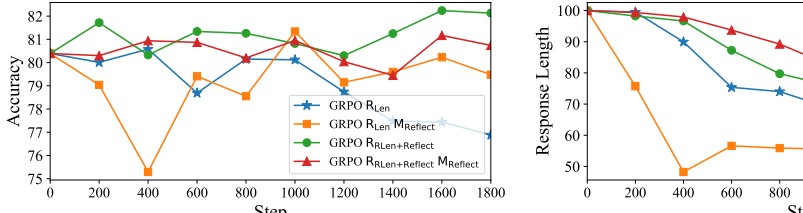

Figure 3: Changes in accuracy and generation length during training on five test sets on average. The x-axis represents the training steps. The left plot shows the average accuracy, and the right plot shows the average token consumption per answer. Abbreviations are aligned with Table 2.

**Mitigating overthinking on easy problems.** LRMs often overthink on easy problems. Our method appropriately reduces reflection on easy problems while maintaining it on difficult ones. While our approach increases the reflection frequency on easier problems compared to $R_{Len}$, it remains lower than that of the original model. Specifically, on the three easiest datasets, we reduce the reflection frequency by 22%, with a 45% efficiency enhancement. Whereas on challenging ones, the reduction in reflection is only 4%, with a 27% efficiency enhancement. This demonstrates that our approach achieves a favorable balance, mitigating overthinking on easy problems while preserving reflection capabilities. Furthermore, we provide additional evidence demonstrating that our method reduces overthinking while preserving reflection capabilities in Appendix B.6.

**Training dynamics analysis.** To provide further analysis, we illustrate the training dynamics in Figure 3. For GRPO $R_{Len}$, after 1,000 steps, when the length falls below 70% of its original value, its performance begins to drop, indicating that the model begins to remove meaningful steps. However, the reflection reward enables a gradual improvement in efficiency with minimal change in accuracy. Besides, the reflection model greatly accelerates length reduction. Although it initially causes severe performance degradation, the targeted penalties that it imposes on overthinking tokens avoid deleting valid reflection tokens, halt further length shortening, and gradually recover performance. After 1,000 steps, it consistently outperforms the baseline in both accuracy and efficiency. Finally, the reflection model's goal remains to penalize overthinking, which is inherently at odds with the reflection reward that indiscriminately encourages reflection. Therefore, their combination achieves a trade-off between efficiency and accuracy, yielding a balance rather than a synergistic gain.

# 6 CONCLUSION AND LIMITATIONS

To enhance the inference efficiency of LRMs without compromising model performance, we propose REflection-Aware online Reinforcement Learning, `REA-RL`, to achieve improved online scaling and better performance retention. Specifically, we introduce a reflection model for efficient scaling, offering sequential revision to augment parallel sampling data generation for online RL. Furthermore, we introduce a reflection reward to better maintain model performance. Experiments show that REA-RL reduces token cost by 36% with no performance loss. Analysis shows that our method is effective by maintaining reflection frequency for challenging problems while appropriately reducing it for easier ones, thus balancing performance and efficiency.

Our paper has the following limitations. First, our approach is only validated on distilled 7B LRMs. Due to the large model size and long training time, we do not perform validation on LRMs pre-trained from scratch, which aligns with prior work (Zhang & Zuo, 2025; Huang et al., 2025; Qu et al., 2025b). Second, the detection method proposed in §3 is based on LLMs. While it outperforms fixed truncation, it cannot guarantee the complete elimination of overthinking. Nevertheless, our goal is to train the reflection model, and as long as it can reduce overthinking to a certain extent, it can effectively shorten the reasoning process during iterative training. Finally, our scaling method introduces approximately 10% additional cost compared to using parallel scaling only. This is due to the use of sequential scaling, which results in poorer parallelism for vllm. However, considering our significant improvement and the cost-free improvement of the reflection reward, this is acceptable.

ETHICS STATEMENT

Our work adheres to the ICLR Code of Ethics and uses publicly available datasets for reproducibility. LLMs may exhibit racial and gender biases, so we strongly recommend users assess potential biases before applying the models in specific contexts. Additionally, due to the difficulty of controlling LLM outputs, users should be cautious of issues arising from hallucinations.

REPRODUCIBILITY STATEMENT

We make our code, configuration files, and evaluation scripts available at the anonymous repository linked in the abstract (`https://github.com/hexuandeng/REA-RL`). All hyperparameters required to reproduce our method are provided in Appendix B.2. The required hardware and runtime are reported in §5.3. The use of large language models is discussed in Appendix B.7.

ACKNOWLEDGMENTS

This work was supported in part by the Guangdong S&T Program (Grant No. 2024B0101050003), Guangdong Basic and Applied Basic Research Foundation (Grant No. 2024A1515011491), and Shenzhen Science and Technology Program (Grant Nos. ZDSYS20230626091203008,KJZD202310230947 00001,KQTD2024072910215406).

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

## A    PROMPTS

We introduce all prompts used in the main text. Parts enclosed in "{}" represent external input.

- **The Prompts for Detecting Overthinking:** These prompts are used in §3.1. We first segment the response into chunks, specifically by splitting on a blank line (i.e., two consecutive newline characters, \n\n). To prevent a single formula from being split across multiple chunks, we merge chunks until the combined chunk ends with a sentence-ending punctuation mark or exceeds 128 tokens. Subsequently, we feed all the chunks into the first prompt for initial detection. To prevent the model from generating excessively long labeling strings, we feed at most 1k tokens at a time. Each chunk subsequently labeled as "Right Result" is then fed into the second prompt for secondary labeling. "With / without Gold Answer" indicates whether the gold answer is included as input.

- **The SFT Prompt for Reflection Model Training:** These prompts constitute the SFT data in §4.2 to train a 7B reflection model. We employ simpler definitions and non-chain-of-thought output to ensure both effectiveness and efficiency. The gold answer is not used as input to ensure usability during training and evaluation.

---

**The First Prompt for Detecting Overthinking with Gold Answer**

\*\*Question:\*\* {Question}
\*\*Gold Answer:\*\* {Answer}
\*\*Response:\*\* {All Chunked Responses}
You are provided with a math Question, a Gold Answer and a model-generated Response. The response is divided into {N} parts. For each part, analyze it and classify it based on its relationship to the provided context. For each part, assign one of the following labels:
   - Reasoning: The part represents the reasoning process that leads to the answer.
   - Right Result: The part is the answer provided by the model, where the model may provide the answer in the middle of its response, and the answer aligns with the Gold Answer.
   - Wrong Result: Same as Right Result, but the answer does not align with the Gold Answer.
For each of the {N} parts, please reply in format:
[1]. Think: [Explanation for label choice]
Label: Reasoning/Right Result/Wrong Result
[2]. Think: [Explanation for label choice]
Label: Reasoning/Right Result/Wrong Result
...

---

**The First Prompt for Detecting Overthinking without Gold Answer**

\*\*Question:\*\* {Question}
\*\*Response:\*\* {All Chunked Responses}
You are provided with a math Question and a model-generated Response. The response is divided into {N} parts. For each part, analyze it and classify it based on its relationship to the provided context. For each part, assign one of the following labels:
   - Reasoning: The part represents the reasoning process that leads to the answer.
   - Right Result: The part is the answer provided by the model, where the model may provide the answer in the middle of its response, and the answer aligns with the Gold Answer.
   - Wrong Result: Same as Right Result, but the answer does not align with the Gold Answer.
For each of the {N} parts, please reply in format:
[1]. Think: [Explanation for label choice]
Label: Reasoning/Right Result/Wrong Result
[2]. Think: [Explanation for label choice]
Label: Reasoning/Right Result/Wrong Result
...

- **The Prompt for Math Evaluation:** We follow Yang et al. (2024b) to design the prompts for math evaluation. For NoThink, we append "Okay, I think I have finished thinking. ¡/think¿" after the prompt to force LRMs to skip reasoning. When forcing the model to complete generation and provide the final answer, we append "¡/think¿ **Final Answer:**" after the thought process and allow the model to continue generating up to 1k tokens. During training, to ensure efficiency, we reduce the budget to 256 tokens.

---

**The Second Prompt for Detecting Overthinking with Gold Answer**

**Question:** {Question}
**Gold Answer:** {Answer}
**Response:** {One Chunked Response}
Evaluate whether the model correctly answered the question. As long as the model provides the correct result, it counts as correct, regardless of format or wording. The response I provided is part of the complete response, so there's no need to include the entire reasoning process. Please judge only if the model has provided the correct answer up to this point. Please reason step by step first after "Reasoning:", then answer only with Yes or No after "Answer:".

---

**The Second Prompt for Detecting Overthinking without Gold Answer**

**Question:** {Question}
**Response:** {One Chunked Response}
Evaluate whether the model has already answered the question. The response I provided is part of the complete response, so there's no need to include the entire reasoning process. Please judge only if the model has provided the answer up to this point. Please reason step by step first after "Reasoning:", then answer only with Yes or No after "Answer:".

---

**The SFT Prompt for Reflection Model Training**

**Question:** {Question}
**Response:** {One Chunked Response}
You are provided with a math Question and a model-generated Response. The response is divided into {N} parts. For each part, analyze it and classify it based on its relationship to the provided context. For each part, assign one of the following labels:
  - Think: The part represents the reasoning process that leads to the answer.
  - Result: The part is the answer provided by the model, where the model may provide the answer in the middle of its response.
For each of the {N} parts, please reply in format:
[1]. Think/Result
[2]. Think/Result
...

---

**The Prompt for Math Evaluation**

Please reason step by step, and put your final answer within \boxed{}.
Question: {Question}

## B FURTHER ANALYSIS

### B.1 CASE STUDY FOR REA-RL

Data filtering (Deng et al., 2023; 2025; Liu et al., 2024b; Rao et al., 2025; Wang et al., 2025) and data synthesis (Deng et al., 2024; Ke et al., 2025; Yuan et al., 2025) are important means of enhancing model capabilities. However, because these techniques are relatively time-consuming, comparatively few studies have explored deploying them in an online setting. To further illustrate how our method works, we provide a case study demonstrating the workflow of online data generation for online RL, containing both parallel sampling and sequential revision, as shown in Figure 4. Specifically, we first perform parallel sampling, obtaining the yellow and red parts. Here, the red part represents the overthinking section, which is detected by the reflection model. Subsequently, we perform sequential revision by removing the red part and allowing the policy model to finish the response with the blue part. In detail, we force the termination of the thinking process by adding a "¡/think¿" token and enforce the generation of the answer by adding "**Final Answer:**" to avoid further redundant reasoning. Based on this process, the blue part is generally much shorter than the red part, thus providing positive cases with less overthinking online.

Since we enforce a limit that no more than half of the tokens in the original response can be removed, and the reflection model cannot always identify the first correct answer, not all additional reflections can be removed. Therefore, the yellow part may also contain some reflection, which also helps REA-RL retain its reflection ability. Additionally, in the second case, the model fails to complete generation within the 8k token budget, resulting in the answer not being formatted as required (within \boxed), and thus marked as incorrect. However, the model has already generated the correct answer, making its response correct after revision. This explains why truncation can sometimes improve performance.

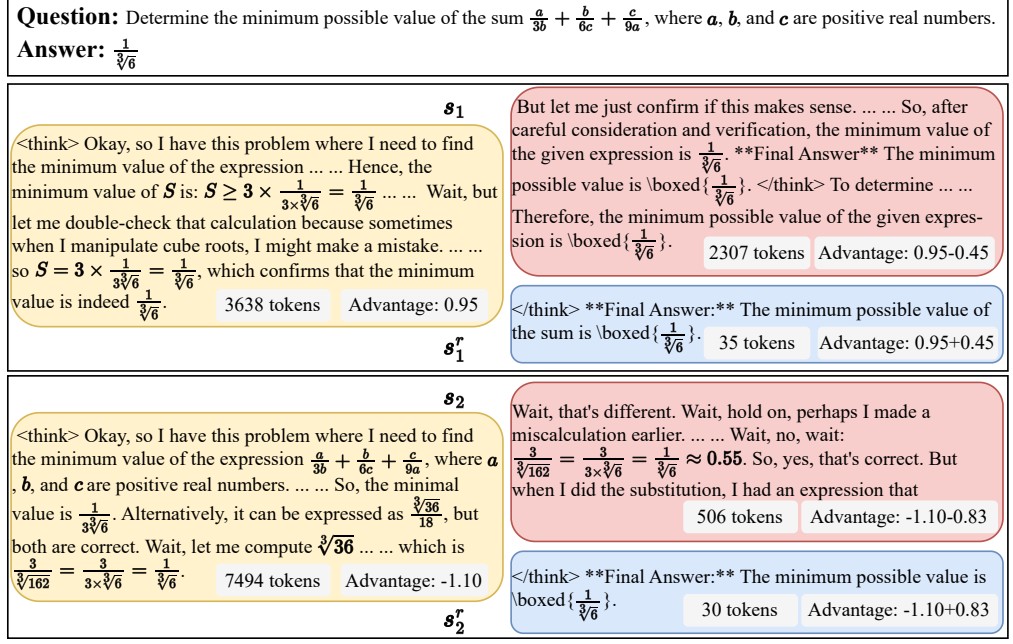

Figure 4: Case study of online data generation in REA-RL. We illustrate how the parallel sampling and sequential revision parts work. Specifically, the yellow, red, and blue parts in this figure correspond to the tokens of the same colors in Figure 2. Since the yellow parts in both completion and revision are identical, they are shown only once in this figure.

### B.2 ADDITIONAL EXPERIMENTAL SETUP

**Common training configuration.** Unless explicitly stated otherwise, we maintain the following experimental settings. We use the DeepScaleR 40k dataset (Luo et al., 2025b) as the training data

| $M_{\text{Reflect}}$ | R1-1.5B | | DeepSeek-7B | | Average | |
|---|---|---|---|---|---|---|
| | $Acc\uparrow$ | $TR\downarrow$ | $Acc\uparrow$ | $TR\downarrow$ | $Acc\uparrow$ | $TR\downarrow$ |
| **Original** | 30.30 | $100_{4844}$ | 49.32 | $100_{3834}$ | 39.81 | $100_{4339}$ |
| **Qwen2.5-7B-Instruct-Weak** | 30.62 | 100.72 | 50.42 | 99.40 | 40.52 | 100.06 |
| **Qwen3-4B-Weak** | 30.48 | 101.96 | 51.60 | 99.48 | **41.04** | 100.72 |
| **Qwen2.5-7B-Instruct-Normal** | 31.40 | 95.11 | 50.12 | 92.96 | 40.76 | 94.04 |
| **Qwen3-4B-Normal** | 31.40 | 96.26 | 50.42 | 94.31 | 40.91 | 95.29 |
| **Qwen2.5-7B-Instruct-Strong** | 30.60 | 89.91 | 48.44 | 87.14 | 39.52 | **88.53** |
| **Qwen3-4B-Strong** | 31.16 | 89.88 | 48.70 | 87.48 | 39.93 | 88.68 |

Table 5: General-QA reflection model evaluation on MMLU-Pro using the truncation-and-force-answer protocol. Other abbreviations are defined in Table 3.

| Method | MMLU | | MMLU-Pro | | SuperGPQA | | Average | |
|---|---|---|---|---|---|---|---|---|
| | $Acc\uparrow$ | $TR\downarrow$ | $Acc\uparrow$ | $TR\downarrow$ | $Acc\uparrow$ | $TR\downarrow$ | $Acc\uparrow$ | $TR\downarrow$ |
| *Budget: 8k* | | | | | | | | |
| **Original** | 40.42 | $100_{1598}$ | 27.44 | $100_{3632}$ | 13.52 | $100_{4441}$ | 27.13 | $100_{3214}$ |
| **GRPO** | 40.36 | 74.55 | 28.10 | 85.77 | 11.98 | 98.13 | 26.81 | 86.15 |
| **GRPO + $M_{\text{Reflect}}$** | 40.94 | 73.85 | 27.88 | 89.98 | 13.56 | 93.22 | **27.46** | **85.68** |
| *Budget: 16k* | | | | | | | | |
| **Original** | 40.98 | $100_{1871}$ | 30.30 | $100_{4844}$ | 16.66 | $100_{6337}$ | 29.31 | $100_{4351}$ |
| **GRPO** | 40.58 | 70.82 | 30.24 | 84.56 | 15.36 | 104.92 | 28.73 | 86.77 |
| **GRPO + $M_{\text{reflect}}$** | 41.30 | 74.77 | 30.48 | 89.66 | 16.86 | 91.68 | **29.55** | **85.37** |

Table 6: General-QA policy training results on general QA. Abbreviations are defined in Table 2.

and DeepSeek-R1-Distill-Qwen-7B (R1-7B) as the base model. We only retain questions for which the model can provide at least one correct answer within 4 samples as the training data, resulting in 30k questions. We train for one epoch for GRPO-based methods and the same number of steps for all other methods and baselines. Each batch contains 16 different questions and 4 different paths generated for each question, with a maximum generation length of 8k. We use a learning rate of $2 \times 10^{-5}$, integrating low-rank adaptation (Hu et al., 2022) with all attention and MLP parameters, setting the LoRA rank $r$ to 16 and alpha to 32. During the data generation process, we follow R1 to use a temperature of 0.6 and a top_p of 0.95. Finally, following Yu et al. (2025), when all generated trajectories yield incorrect answers, we skip these instances to avoid unintended optimization.

### B.3 GENERALIZATION RESULTS ON BROADER DOMAINS AND MODEL SCALES

To address the concern that our main experiments focus on the math domain, we add experiments on general QA reasoning and smaller LRMs following prior evaluation setups.

**Models and reflection model selection.** We intentionally use smaller policy models to verify generality in model scales. Specifically, we report results on DeepSeek-R1-Distill-Qwen-1.5B (R1-1.5B). For the general QA setting, we also train a new reflection model: we generate SFT data with R1-1.5B as the trajectory generator and Qwen3-32B (Yang et al., 2025) as the reviser, mix these QA-domain data with the math-domain data, and train reflection models based on Qwen2.5-7B-Instruct and Qwen3-4B.

**General QA domain dataset.** Concretely, we evaluate on MMLU (Hendrycks et al., 2021a), MMLU-Pro (Wang et al., 2024), and SuperGPQA (Team et al., 2025), covering different difficulty levels. Based on the dataset sizes, we select 20k examples from MMLU, 5k from MMLU-Pro, and 15k from SuperGPQA to construct a 40k mixed training set, and we use 5k examples from each dataset as test sets. Since some QA questions do not require deep reasoning, we first filter out the 15% of questions with the shortest paths (generated by R1-1.5B), and then sample the required number of examples. Finally, for the training set, we remove questions that R1-1.5B cannot solve within 4 paths, resulting in a 30k-example training set.

| Method | GSM8K | | Math500 | | GaoKao23 | | Amc23 | | Aime | | Average | |
|---|---|---|---|---|---|---|---|---|---|---|---|---|
| | Acc↑ | TR↓ | Acc↑ | TR↓ | Acc↑ | TR↓ | Acc↑ | TR↓ | Acc↑ | TR↓ | Acc↑ | TR↓ |
| *Budget: 8k* | | | | | | | | | | | | |
| Original | 75.51 | $100_{615}$ | 67.80 | $100_{4137}$ | 65.97 | $100_{3336}$ | 52.81 | $100_{5412}$ | 18.33 | $100_{7581}$ | 56.08 | $100_{4216}$ |
| GRPO | 74.83 | 58.54 | 77.40 | 52.28 | 63.64 | 65.14 | 60.62 | 72.60 | 22.92 | 86.26 | 59.88 | **66.96** |
| GRPO + $M_{Reflect}$ | 72.33 | 60.81 | 80.00 | 55.14 | 64.68 | 67.51 | 65.31 | 73.50 | 24.58 | 88.83 | **61.38** | 69.16 |
| *Budget: 16k* | | | | | | | | | | | | |
| Original | 75.59 | $100_{622}$ | 72.00 | $100_{5652}$ | 69.61 | $100_{4327}$ | 57.81 | $100_{8283}$ | 21.25 | $100_{13358}$ | 59.25 | $100_{6448}$ |
| GRPO | 74.83 | 58.84 | 78.60 | 46.16 | 65.71 | 59.88 | 63.75 | 60.69 | 24.58 | 79.45 | 61.49 | **61.00** |
| GRPO + $M_{reflect}$ | 72.33 | 60.13 | 81.60 | 48.73 | 68.05 | 59.51 | 69.06 | 62.77 | 28.75 | 75.58 | **63.96** | 61.34 |

Table 7: Additional math-domain results on R1-1.5B. Abbreviations are defined in Table 2.

**Policy training setup.** We use the same training configuration as in the main text. For general QA, the prompt still uses the same setting as in the main text, i.e., "Please reason step by step, and put your final answer within \boxed{}. Question: {Question}." We only replace the question with the concatenation of the question and choices for General QA domain. After training, since the dataset is large enough, we generate one path for each question and report the average results, with generation hyperparameters consistent with the main text. For the additional math-domain experiments on R1-1.5B, we follow the same training and generation configuration as in the main experiment to ensure a controlled comparison.

**Reflection model evaluation results on general QA.** Under the same truncation-and-force-answer evaluation protocol (tested on MMLU-Pro), the trained reflection models can effectively truncate trajectories with comparable performance across 7B and 4B reflection models, as shown in Table 5. Across reflection model sizes, we observe a consistent trade-off controlled by the truncation strength: the *Normal* setting noticeably reduces TR while maintaining or slightly improving average accuracy, whereas *Strong* further reduces TR but can introduce a larger accuracy drop. Overall, these results indicate that the reflection model transfers to the QA domain under the same evaluation protocol, and that smaller reflection models (4B) can closely match the 7B reflection model in both accuracy and truncation behavior. Therefore, we use the 4B model as the reflection model for general QA in the following experiments.

**Policy training results on general QA and math.** Table 6 reports policy training results on general QA with R1-1.5B. Under both 8k and 16k budgets, introducing the reflection model yields stable improvements in average accuracy while maintaining a comparable average token ratio. Table 7 further reports additional experiments in the math domain on R1-1.5B, using the same training and generation configuration as in the main experiments: under both budgets, our approach improves average accuracy compared to GRPO while achieving comparable average token ratios. Taken together, these results demonstrate that the approach generalizes to smaller models across both math and general QA domains.

### B.4 ABLATION STUDY OF REWARD REFINEMENT

To further validate our proposed reward refinement scheme, including the improvements to reflection reward and length reward, the results are presented in Table 8.

**Refined length accelerates response shortening.** We evaluate the performance of the length reward before and after optimization (Len vs. RLen) when using only the accuracy reward and when using both the accuracy reward and the reflection reward. The results demonstrate that RLen can significantly reduce the number of tokens used while maintaining or even improving performance. We attribute this to the removal of positive signals for incorrect answers, which avoids encouraging erroneous responses and reduces input noise, thereby accelerating convergence.

**Reflection reward improves performance but reduces efficiency.** Across all experiments, we observe an average performance gain of 4.26 compared to the scenarios without the reflection reward. However, this improvement comes at the cost of an average 23.26 reduction in the truncation ratio. Nevertheless, we believe that maintaining performance is a more critical objective than shortening

| | Method | GSM8K | | Math500 | | Gaokao23 | | Amc23 | | Aime24 | | Average | |
|---|---|---|---|---|---|---|---|---|---|---|---|---|---|
| | | $Acc\uparrow$ | $TR\downarrow$ | $Acc\uparrow$ | $TR\downarrow$ | $Acc\uparrow$ | $TR\downarrow$ | $Acc\uparrow$ | $TR\downarrow$ | $Acc\uparrow$ | $TR\downarrow$ | $Acc\uparrow$ | $TR\downarrow$ |
| **Budget: 8k** | **Original** | 91.58 | $100_{1312}$ | 88.40 | $100_{3389}$ | 77.40 | $100_{3270}$ | 77.50 | $100_{4746}$ | 40.83 | $100_{6815}$ | 75.14 | $100_{3906}$ |
| | **GRPO $R_{Len}$** | 85.97 | 32.01 | 87.60 | 59.04 | 70.91 | 58.84 | 85.62 | 71.43 | 45.42 | 87.42 | 75.10 | 61.75 |
| | **GRPO $R_{RLen}$** | 86.05 | 29.42 | 88.20 | 50.19 | 68.83 | 51.10 | 86.88 | 63.27 | 48.75 | 84.92 | 75.74 | 55.78 |
| | **GRPO $R_{Len+Reflect}$** | 92.95 | 89.63 | 91.60 | 81.74 | 81.30 | 84.68 | 83.12 | 87.00 | 44.58 | 95.52 | 78.71 | 87.71 |
| | w/ $D_{0.1}$ | 92.34 | 80.26 | 91.60 | 76.54 | 79.22 | 78.87 | 84.06 | 82.64 | 40.00 | 93.25 | 77.44 | 82.31 |
| | w/ $D_{0.4}$ | 93.25 | 108.46 | 88.00 | 93.24 | 80.52 | 96.51 | 81.88 | 92.82 | 37.50 | 100.18 | 76.23 | 98.24 |
| | **GRPO $R_{RLen+Reflect}$** | 92.72 | 67.07 | 91.40 | 69.52 | 80.52 | 72.54 | 85.62 | 75.64 | 47.92 | 91.37 | **79.64** | **75.23** |
| **Budget: 16k** | **Original** | 91.66 | $100_{1344}$ | 92.00 | $100_{3893}$ | 81.82 | $100_{3785}$ | 88.12 | $100_{5840}$ | 48.33 | $100_{10460}$ | 80.39 | $100_{5064}$ |
| | **GRPO $R_{Len}$** | 85.97 | 31.25 | 88.40 | 56.43 | 72.21 | 55.88 | 87.81 | 65.65 | 50.00 | 76.96 | 76.88 | 57.23 |
| | **GRPO $R_{RLen}$** | 86.05 | 28.72 | 89.20 | 46.62 | 69.35 | 47.03 | 88.12 | 56.59 | 54.17 | 75.40 | 77.38 | 50.87 |
| | **GRPO $R_{Len+Reflect}$** | 92.95 | 88.10 | 93.20 | 79.58 | 82.86 | 79.39 | 87.50 | 83.30 | 51.67 | 89.54 | 81.64 | 83.98 |
| | w/ $D_{0.1}$ | 92.34 | 78.65 | 93.20 | 73.67 | 81.56 | 77.31 | 89.06 | 80.62 | 47.92 | 91.85 | 80.82 | 80.42 |
| | w/ $D_{0.4}$ | 93.40 | 107.59 | 91.60 | 93.68 | 83.90 | 92.26 | 89.69 | 92.47 | 46.25 | 101.19 | 80.97 | 97.44 |
| | **GRPO $R_{RLen+Reflect}$** | 92.72 | 65.48 | 92.80 | 66.71 | 81.82 | 68.27 | 88.75 | 72.17 | 54.58 | 86.21 | **82.13** | **71.77** |

Table 8: Results of the ablation study. The table presents two sets of experiments using $R_{Len}$ and $R_{RLen}$ to demonstrate the effectiveness of our length reward optimization, as well as an ablation study on the hyperparameters of the reflection reward. "w/ $D_{0.1}$" and "w/ $D_{0.4}$" are defined in Equation 3, representing the use of the 0.1 and 0.4 quantiles of the reflection density for the reflection reward, respectively. Other abbreviations are defined in Table 2.

the response. Only with the addition of the reflection reward can we achieve a reduction in model response length without sacrificing model performance, and we anticipate further shortening of the response with continued training, as illustrated in Figure 3. Conversely, continuing training with only the length reward may lead to a further decline in accuracy.

**Quantile hyperparameter settings in reflection reward.** In the reflection reward, we utilize the 0.2 quantile of the reflection density during training, i.e., $D_{0.2}$. To investigate the impact of other quantiles on the experimental results, we further explore the effects of the 0.1 and 0.4 quantiles. Specifically, $D_{0.1} = 1/299$, $D_{0.2} = 1/225$, and $D_{0.4} = 1/157$, where the denominator represents the number of tokens between reflection tokens at that quantile. The results indicate that as the quantile increases, the penalty for reflection density becomes more severe, leading to longer responses. However, its impact on performance is relatively small, especially with a sufficient budget. This demonstrates that the effectiveness of the reflection reward stems from discouraging non-reflection behavior rather than unconditionally encouraging reflection.

### B.5 TOKEN COST ANALYSIS AND TRAINING-TIME OVERHEAD

**Token consumption and cost.** We measure token usage per path during training in our main experiments. For parallel sampling, the average input length is 94 tokens and the average output length is 3605 tokens. For sequential revision, the input is 6678 tokens and the output is 319 tokens in total. Given that input tokens are typically at most 1/5 the price of output tokens, converting output tokens into input tokens can substantially reduce cost under large-scale generation. Using publicly available API pricing as an example, for GPT-5, the input price is $1.25 per 1M tokens ($0.125 for cached input), and the output price is $10.00 per 1M tokens. Under this pricing, parallel sampling for 1k examples costs $36.17, whereas sequential revision costs $7.78.

Further, we measure small-scale vLLM inference time on a per-batch basis, which aligns with our training setup. On a single A800 GPU, generating one batch with 4 paths takes 158s; sequential revision takes 47s for labeling plus 2s for answer generation (49s total), which is less than one third of the traditional rollout. In our online pipeline, these stages run sequentially per batch (158 + 49 = 207s), which can be slower than pure parallel sampling due to reduced parallelism, i.e., sampling 8 paths in parallel takes 192s. Therefore, in practice we spend more time on training. In our setup on 3 A800 80G GPUs, sampling 4 paths takes 80h, sampling 8 paths takes 110h, and incorporating sequential revision takes 120h. However, at larger generation scale this loss of parallelism is mitigated, bringing the final cost closer to the price ratio estimated by the above API calls.

**Heuristic truncation at reflection tokens.** Although introducing a reflection model incurs additional latency, we find it necessary, as heuristic truncation is unlikely to achieve consistently strong results. We add a baseline that truncates the reasoning trace at the first (or fifth) sentence that begins

| Method | GSM8K Acc↑ | GSM8K TR↓ | Math500 Acc↑ | Math500 TR↓ | Gaokao23 Acc↑ | Gaokao23 TR↓ | Amc23 Acc↑ | Amc23 TR↓ | Aime24 Acc↑ | Aime24 TR↓ | Average Acc↑ | Average TR↓ |
|---|---|---|---|---|---|---|---|---|---|---|---|---|
| Original | 91.66 | $100_{1344}$ | 92.00 | $100_{3893}$ | 81.82 | $100_{3785}$ | 88.12 | $100_{5840}$ | 48.33 | $100_{10460}$ | 80.39 | $100_{5064}$ |
| First | 69.37 | 29.84 | 49.80 | 14.98 | 43.38 | 15.67 | 29.06 | 11.92 | 2.08 | 5.87 | 38.74 | 15.66 |
| Fifth | 89.69 | 68.01 | 70.80 | 40.41 | 60.52 | 38.57 | 55.00 | 29.76 | 16.25 | 15.50 | 58.45 | 38.45 |
| 32B Revise | 89.46 | 62.43 | 93.20 | 71.85 | 80.52 | 70.41 | 89.06 | 79.95 | 50.83 | 93.59 | 80.61 | 75.65 |
| GRPO $R_{RLen+Reflect}$ $M_{Reflect}$ | 89.99 | 52.08 | 91.00 | 60.78 | 82.08 | 55.85 | 89.38 | 67.76 | 51.25 | 81.09 | 80.74 | 63.51 |

Table 9: Heuristic truncation baselines vs. learned truncation model. Other abbreviations are defined in Table 3.

| | Method | GSM8K Acc↑ | GSM8K TR↓ | Math500 Acc↑ | Math500 TR↓ | Gaokao23 Acc↑ | Gaokao23 TR↓ | Amc23 Acc↑ | Amc23 TR↓ | Aime24 Acc↑ | Aime24 TR↓ | Average Acc↑ | Average TR↓ |
|---|---|---|---|---|---|---|---|---|---|---|---|---|---|
| **Budget: 16k** | Original | 91.66 | $100_{1344}$ | 92.00 | $100_{3893}$ | 81.82 | $100_{3785}$ | 88.12 | $100_{5840}$ | 48.33 | $100_{10460}$ | 80.39 | $100_{5064}$ |
| | Model Reflect | 89.46 | 62.43 | 93.20 | 71.85 | 80.52 | 70.41 | 89.06 | 79.95 | 50.83 | 93.59 | 80.61 | 75.65 |
| | + Gold | 88.78 | 54.09 | 92.00 | 57.80 | 79.48 | 59.71 | 87.19 | 67.71 | 51.67 | 92.31 | 79.82 | 66.32 |
| | GRPO $R_{RLen+Reflect}$ | 92.72 | $100_{880}$ | 92.80 | $100_{2597}$ | 81.82 | $100_{2584}$ | 88.75 | $100_{4215}$ | 54.58 | $100_{9018}$ | **82.13** | $100_{3859}$ |
| | Model Reflect | 92.42 | 72.73 | 93.00 | 78.17 | 80.26 | 75.46 | 87.19 | 84.01 | 54.58 | 94.72 | 81.49 | 81.02 |
| | + Gold | 91.74 | 68.75 | 91.60 | 69.77 | 79.74 | 70.51 | 85.94 | 77.27 | 54.58 | 93.70 | 80.72 | 76.00 |
| | GRPO $R_{Len}$ $M_{Reflect}$ | 89.23 | $100_{496}$ | 92.40 | $100_{1756}$ | 79.74 | $100_{1798}$ | 88.12 | $100_{3273}$ | 47.92 | $100_{7931}$ | 79.48 | $100_{3051}$ |
| | Model Reflect | 88.02 | 90.73 | 91.40 | 92.54 | 79.22 | 92.99 | 88.75 | 93.92 | 45.42 | 95.51 | 78.56 | 93.14 |
| | + Gold | 87.72 | 88.91 | 89.80 | 84.85 | 78.96 | 88.65 | 86.56 | 90.53 | 46.25 | 95.21 | 77.86 | 89.63 |
| | GRPO $R_{RLen+Reflect}$ $M_{Reflect}$ | 89.99 | $100_{700}$ | 91.00 | $100_{2366}$ | 82.08 | $100_{2114}$ | 89.38 | $100_{3957}$ | 51.25 | $100_{8482}$ | 80.74 | $100_{3524}$ |
| | Model Reflect | 90.30 | 86.29 | 91.60 | 88.08 | 80.78 | 88.69 | 88.75 | 90.90 | 50.42 | 96.24 | 80.37 | 90.04 |
| | + Gold | 89.46 | 80.86 | 90.40 | 80.85 | 79.74 | 82.64 | 87.19 | 88.10 | 50.83 | 95.48 | 79.52 | 85.59 |

Table 10: Results of the overthinking analysis. Each group of results is divided into three rows. The first row shows the performance of the original model or the trained model directly. The following two rows represent the results of shortening the responses of the first-row model using different methods to remove overthinking. "Model Reflect" and "+ Gold" are defined in Table 3, representing revision using the 32B model. Other abbreviations are defined in Table 2.

with a reflection token, and then forces final-answer generation. Results are shown in Table 9. Truncating at the first occurrence is too aggressive and causes a large accuracy drop, because reflection can occur during intermediate reasoning rather than only after finishing. Truncating at the fifth occurrence improves results on easier datasets but still yields substantial drops on harder datasets, indicating that the optimal truncation strength is difficulty-dependent and hard to tune; this supports the necessity of a learned reflection model.

## B.6 OVERTHINKING ANALYSIS AFTER TRAINING

**Significantly lower overthinking.** To verify that our method alleviates the overthinking issue, we employ the same evaluation approach as in §3, which uses an LLM to identify overthinking tokens and remove them with model reflection. The results are presented in Table 10. Compared to directly truncating the response from the original model, our approach exhibits a considerably lower truncation ratio for overthinking tokens, especially for methods employing our reflection model for training. This demonstrates that REA-RL effectively mitigates the issue of overthinking.

**Preserving reflection capability.** While our method reduces the truncation ratio, often indicating less additional reflection, it does not eliminate it entirely. A certain proportion of truncation is still maintained across our methods, particularly in experiments utilizing our reflection reward. This indicates that our method still retains a certain token budget for the final reflection, thus proving that REA-RL preserves the reflection capability.

## B.7 THE USE OF LARGE LANGUAGE MODELS

As a general-purpose assist tool, large language models are used only to aid writing (e.g., grammar, clarity, and phrasing). They are not used to design methods, generate ideas, run experiments, or create results. All technical content and conclusions are written and verified by the authors.

