# OpenReview forum: "REA-RL: Reflection-Aware Online Reinforcement Learning for Efficient Reasoning"
_ICLR.cc/2026/Conference — ICLR 2026 Poster_

### Official Review · Reviewer_1pmc · 2025-10-15

**Soundness:** 2
**Presentation:** 4
**Contribution:** 3
**Rating:** 6
**Confidence:** 3

**Summary:**

This paper proposes REA-RL, a reflection-aware online reinforcement learning framework aimed at improving the efficiency of reasoning in Large Reasoning Models (LRMs) without sacrificing accuracy. The work addresses the challenge of overthinking in chain-of-thought (CoT) reasoning, where models often generate excessively long, redundant reasoning steps that increase inference cost with limited benefit.

REA-RL introduces two key components:

Reflection Model: A lightweight model trained to identify the earliest point in a reasoning trace where the correct answer appears. Based on this, the model trims subsequent “overthinking” tokens and generates a concise revision. This enables efficient sequential revision that complements parallel sampling in online RL.

Reflection Reward: A new reward function based on the density of reflective tokens (e.g., "wait", "but") that penalizes non-reflective outputs, which often result from using only length-based rewards.

The proposed framework is implemented within a Grouped Relative Policy Optimization (GRPO) pipeline and combines both the original and revised responses for policy updates. Experiments on several math reasoning benchmarks (e.g., GSM8K, Math500, AMC23) show that REA-RL can reduce token usage by 36% while maintaining or even improving accuracy. The analysis further demonstrates that the system preserves reflection on hard problems and reduces unnecessary reflection on easier ones, achieving a better balance between efficiency and performance than prior approaches.

**Strengths:**

- The paper introduces a reflection-aware revision framework (REA-RL) combining a lightweight reflection model and reflection reward, achieving a 36% inference efficiency gain with no performance loss.
- The detection method is principled, with a clear revision boundary strategy and efficient implementation.
- The reward shaping component addresses the typical degeneration from using length-only rewards in online RL.
- Experiments are thorough and well-structured, with both ablations and scaling comparisons.

**Weaknesses:**

- Limited to 7B distilled models need 32Bmodel for guidance; lacks generalization validation to pre-trained or larger LLMs.
- The reflection detection relies on LLM-based heuristics, which may not scale or generalize across domains.
- Added complexity in training pipeline (sequential revision) increases compute by ~10%.

**Questions:**

Truncation is somehow brutal? what if answeer first, and explain later.
when calculate  overthink signals ,wait, but , alternativly, is somehow way too heuristic?
and chose 20th  for punishment, do we have experiment to support this parameter choose?

---

> ### Author Response · Authors · 2025-11-27
> **Official Comment to Reviewer 1pmc (1/2)**
>
> Thank you very much for your timely response and for your **positive assessment of our replies**. As you may have seen, some of our replies to other reviewers overlap with your comments, especially our replies to **Reviewer xPcf’s Q1–Q3**, which closely match the **three weaknesses** you raised. To avoid making you re-read the same material, we first briefly summarize those results and how they address your three weaknesses, and then focus on the parts of your questions that are not covered elsewhere.
>
> > Summary of our response to the three weaknesses.
>
> In our response to Reviewer xPcf’s Q1–Q3, we added the following **experiments and analyses**, which also **mitigate your concerns**:
>
> - Because GRPO training is time- and resource-intensive, it is difficult for us to include a 32B model. Instead, we added **generalization validation on pre-trained LLMs of different sizes**: **SmolLM3-3B**, a non-DeepSeek, non-Qwen model series pre-trained from scratch and released by Hugging Face on September 10 (69k downloads), and **DeepSeek-R1-Distill-Qwen-1.5B**. The results on these models validate the generalization of our method to pre-trained models and different LLM sizes.
>
> - We trained a **reflection model for the general QA domain**, covering both **7B and 4B variants**. This demonstrates that the **reflection detection method can generalize across domains**, and that smaller reflection models can achieve performance comparable to the 7B reflection model.
>
> - Although the added complexity in the training pipeline increases compute by roughly 10%, the performance gains are substantial enough to justify this extra cost (a 3% performance improvement and 4% shorter outputs compared to sampling 8 paths). Besides, our approach generates far **fewer output tokens** than a full rollout; the slower training speed is mainly due to **reduced parallelism**. In API-based or large-scale training deployments, we expect our method to be **as fast as**, or even **faster than**, the baseline.
>
> > Q1: Truncation is somehow brutal? What if answer first, and explain later?
>
> Thank you for the suggestion; this is indeed a very insightful point. Our truncation scheme is designed specifically for **decoder-only models** under the next-token prediction paradigm, where each token is generated conditioned only on previous tokens. Therefore, if the model answers first and explains later, we consider the later explanation as not contributing to the answer: any explanation that appears **after** the final answer is treated as overthinking, while the explanation **before** the answer is sufficient to obtain the correct answer and is treated as effective reasoning.
>
> As a result, this truncation scheme is intended to apply only to mainstream decoder-only LLMs. We will clarify this limitation more explicitly in the paper. Thanks again for your insightful question.

---

> > ### Author Response · Authors · 2025-11-27
> > **Official Comment to Reviewer 1pmc (2/2)**
> >
> > > Q2: When calculate overthink signals, wait, but, alternativly, is somehow way too heuristic?
> >
> > Thank you for the suggestion; this is a very important question. We use these words because reflective keywords can act as **trigger words** that cause a model to start reflecting once they appear, both in fine-tuned and pre-trained models. In practice, increasing or decreasing the frequency of these keywords effectively increases or decreases the frequency of reflection.
> >
> > [1] shows that simply inserting reflective keywords can elicit reflection even in pre-trained models, likely because they frequently co-occur with reflective reasoning in the pre-training corpus. [2] further finds that the hidden states preceding reflective keywords differ systematically from other hidden states, and that linearly intervening along the difference between these states can encourage or suppress reflection. Together, these results show that these keywords are tightly coupled with reflection, because they are highly correlated with reflective behavior in the pre-training data and thus have the potential to transfer across diverse models.
> >
> > Building on this observation, several decoding methods explicitly modify the probabilities of reflective keywords to control how often models reflect [3–6], and their success provides further evidence for the strong link between reflective tokens and reflection frequency. Finally, multiple works analyze these keywords across a variety of reasoning models [7–9], consistently reporting the presence of such tokens. These reflective keywords have been widely shown to appear frequently in pre-training corpora and to correlate strongly with reflection in many reasoning models. This makes them a practical and empirically validated choice, rather than an ad-hoc heuristic.
> >
> > [1] Rethinking Reflection in Pre-Training, arXiv 2025. 35 citations.
> >
> > [2] From Emergence to Control: Probing and Modulating Self-Reflection in Language Models, arXiv 2025.
> >
> > [3] Efficient Reasoning for Large Reasoning Language Models via Certainty-Guided Reflection Suppression, AAAI 2026.
> >
> > [4] Speculative Thinking: Enhancing Small-Model Reasoning with Large Model Guidance at Inference Time, COLM 2025.
> >
> > [5] Thoughts Are All Over the Place: On the Underthinking of o1-Like LLMs, arXiv 2025. 83 citations.
> >
> > [6] CyclicReflex: Improving Reasoning Models via Cyclical Reflection Token Scheduling, arXiv 2025.
> >
> > [7] Missing Premise Exacerbates Overthinking: Are Reasoning Models Losing Critical Thinking Skill? COLM 2025.
> >
> > [8] Exploring and Exploiting the Inherent Efficiency within Large Reasoning Models for Self-Guided Efficiency Enhancement, arXiv 2025.
> >
> > [9] RLVR vs. Distillation: Understanding Accuracy and Capability in LLM Mathematical Reasoning, arXiv 2025.
> >
> > > Q3: Chose 20th for punishment, do we have experiment to support this parameter choose?
> >
> > Thank you for your feedback. We have conducted ablation study on this parameter, as shown in Table 5 in the appendix, which demonstrate its robustness, and we copy the average results in the following table. The results indicate that the accuracy is mildly affected by the choice of quantile; only a large quantile (e.g., 0.4) leads to longer responses. The difference between using 0.1 and 0.2 is negligible: a quantile of 0.1 yields slightly lower accuracy but slightly higher efficiency. This confirms that the quantile is not a hyperparameter that requires careful tuning.
> >
> > | Method  | Avg Acc↑  | Avg TR↓ |
> > |---------------------------------------------|---------------------:|---------------------:|
> > | **Budget: 8k** |
> > | GRPO ${R}_{Len+Reflect}$  | 78.71 | 87.71 |
> > | w/ $D_{0.1}$| 77.44 | 82.31 |
> > | w/ $D_{0.4}$| 76.23 | 98.24 |
> > | **Budget: 16k** |
> > | GRPO ${R}_{Len+Reflect}$  | 81.64 | 83.98 |
> > | w/ $D_{0.1}$| 80.82 | 80.42 |
> > | w/ $D_{0.4}$| 80.97 | 97.44 |
> >
> > Regarding how to choose the quantile, our objective is to penalize the model only for an **excessively low** reflection frequency. For these almost non-reflective patterns, we want the model to produce at least some **trigger words** (e.g., wait, but) to initiate the self-reflection process. We therefore recommend selecting a **low quantile (≤ 0.2)** to achieve good results without detailed hyperparameter search, with its exact value depending on the desired trade-off between performance and efficiency: within this small range, a larger quantile may correspond to slightly higher accuracy but lower efficiency.

---

### Official Review · Reviewer_homH · 2025-10-31

**Soundness:** 4
**Presentation:** 3
**Contribution:** 3
**Rating:** 8
**Confidence:** 4

**Summary:**

This paper introduces REA-RL, an online reinforcement learning framework designed to mitigate overthinking in reasoning language models. REA-RL employs a reflection model to detect and truncate redundant reasoning segments, producing revised trajectories for policy learning. It further incorporates a reflection reward to encourage appropriate self-correction and a refined length reward that promotes concise reasoning only when answers are correct. With this method, REA-RL achieves up to 36% shorter reasoning traces without accuracy degradation across multiple reasoning benchmarks, demonstrating more efficient and balanced reasoning behavior.

**Strengths:**

1. The paper presents an original and well-motivated idea of integrating reflection awareness into reinforcement learning to mitigate overthinking in reasoning models.
2. The approach is clearly explained, with intuitive motivation, and supporting experiments and ablations.
3. Experimental results are consistent and persuasive, demonstrating significant reductions in reasoning length while maintaining accuracy.

**Weaknesses:**

1. The revision model design seems questionable. The paper reports that using the revision model alone achieves even better results than revision model + gold answer, which is counter-intuitive. Since matching the correct answer is not a difficult task, this suggests that the revision mechanism may not be well aligned with correctness or may overfit to surface reflection patterns. Clarifying why this happens would strengthen the paper.
2. The reflection reward based on reflection-token density might lead to reward hacking. Models could insert reflective keywords (“wait”, “check”, “however”) without performing genuine self-correction. A more semantic or context-sensitive reflection metric could better ensure that the reward encourages meaningful reflection rather than stylistic mimicry.

**Questions:**

I’m curious about the training setup: the paper mentions that both the original and revised responses are used together during online updates. Have the authors tried using only the revised responses for training? It would be interesting to see whether excluding the original (possibly overthinking) trajectories leads to more stable learning or better efficiency–accuracy trade-offs.

---

> ### Author Response · Authors · 2025-11-27
> **Official Comment to Reviewer homH (1/4)**
>
> > W1: The revision model design seems questionable. The paper reports that using the revision model alone achieves even better results than revision model + gold answer, which is counter-intuitive. Since matching the correct answer is not a difficult task, this suggests that the revision mechanism may not be well aligned with correctness or may overfit to surface reflection patterns. Clarifying why this happens would strengthen the paper.
>
> Thank you for your feedback; this is a very insightful question. In fact, the gap between the *with-answer* and *without-answer* settings is smaller than it may appear. Under the truncation setup, when the original trajectory fails to solve the problem, truncating at any position almost never recovers the correct answer, so truncation has little effect on the outcome. When the original trajectory solves the problem, the correct answer already appears at the end of that trajectory. In this case, both prompting variants have access to the correct answer. Thus, the two prompts contain essentially equivalent information. The only difference is that in the *without-answer* setting, the model cannot be sure whether the answer it sees is correct, so it tends to prune less aggressively.
>
> Moreover, in the Table 1 results, the *with-answer* setting is not strictly worse: although its accuracy is slightly lower, it produces shorter reasoning and is therefore more efficient. When the model is provided with the gold answer, it tends to adopt a more aggressive pruning strategy. Under our truncation setup, a shorter trajectory strictly contains less information than a longer one, so its performance can hardly surpass that of the without-answer setting. Nevertheless, even under this disadvantage, on AIME’24, the with-answer setting achieves both higher accuracy and a lower token ratio, suggesting that it has greater potential to identify good truncation positions.
>
> Finally, more aggressive pruning sometimes leads to truncation at an unnatural location that makes it harder for the model to produce a natural final answer, which can lead to worse results. We illustrate this with two examples below.
>
> * **Case 1.**
>   * Problem: Joe has \\$50 to buy an outfit for his new field trip. There is a 30% off sale at the clothing store. The shirt he picks out has a price of \\$25. He also picks out a pair of shorts for $35. Assuming that sales tax is included, how much money will Joe have left after the purchase?
>   * Gold answer: 8.
>   * Truncation position in the **with-answer setting**: Alternatively, maybe the problem is expecting me to calculate the total without tax and just subtract that from \\$50, which would be \\$60 - \\$18 = \\$42, so \\$50 - \\$42 = \\$8. But that would ignore the tax, which the question says is included. (Sentence No. 36)
>   * Final answer in the with-answer setting: Joe will have $\boxed{5.90}$ left after the purchase.
>   * Truncation position in the **without-answer setting**: Wait, maybe the problem is expecting me to calculate the total cost, apply the discount, and then subtract from $50, but not worrying about tax. So, \\$60 - \\$18 = \\$42, \\$50 - \\$42 = \\$8. (Sentence No. 60)
>   * Final answer in the without-answer setting: Joe will have $\boxed{8}$ dollars left after the purchase.
>
> From this case, we see that by sentence 36, the model has already obtained the correct answer 8, but then raises an incorrect concern that tax has not been accounted for and subtracts an additional amount (assuming a 5% tax), which leads to the wrong final answer. In the *without-answer* setting, the trajectory is truncated later, after the model has resolved this spurious concern, and it produces the correct answer.

---

> ### Author Response · Authors · 2025-11-27
> **Official Comment to Reviewer homH (2/4)**
>
> * **Case 2.**
>   * Problem: Kayla is having her birthday party at a movie theater. The fee to rent the theater is \\$125 for a party of 20, plus \\$6 for each additional guest. Kayla invited her 25 classmates and the 7 girls in her dance class, as well as 13 family members. Only 4 people said they could not come. How much will the party cost?
>   * Gold answer: 251.
>   * Truncation position in the **with-answer setting**: Wait, perhaps the flat fee is for 20 people, which is 20 guests plus Kayla. So, if she invites 45 guests, but 4 can't come, the number of guests attending is 45 - 4 = 41. So, the number of additional guests beyond the 20-person flat fee is 41 - 20 = 21. So, 21 additional guests. Then, the cost would be \\$125 + 21 * \\$6.” (Sentence No. 13)
>   * Final answer in the with-answer setting: The total cost of the party will be $\boxed{281}$ dollars.
>   * Truncation position in the **without-answer setting**: Alternatively, maybe the flat fee is for 20 guests, not including Kayla. So, if Kayla is hosting, then the 20 is for guests, and the flat fee is just for the theater. So, if she's inviting 45 guests, and 4 can't come, the number of guests attending is 41. Then, beyond the 20 guests, there are 41 - 20 = 21 additional guests. So, cost would be 125 + 21*6 = 251. (Sentence No. 19)
>   * Final answer in the without-answer setting: The total cost of Kayla's party will be $\boxed{251}$ dollars.
>
> Here, by sentence 13, the trajectory already implicitly contains the correct reasoning structure, but the computation is not explicitly completed, leading to an incorrect numerical answer. Only when we show the gold answer do we feel “safe” truncating this early. At sentence 19, the model finishes the computation and obtains the correct result. This shows that even a seemingly good truncation position can still produce an incorrect final answer, and this issue is more severe in the with-answer setting. Thanks again for the insightful question.
>
> > W2: The reflection reward based on reflection-token density might lead to reward hacking. Models could insert reflective keywords (“wait”, “check”, “however”) without performing genuine self-correction. A more semantic or context-sensitive reflection metric could better ensure that the reward encourages meaningful reflection rather than stylistic mimicry.
>
> Thank you for the comment; this is a good suggestion. However, we believe that using keyword frequency is already sufficient to capture the frequency of reflection, and using it as a reward can achieve good performance without reward hacking. Reflective keywords can serve as trigger words that prompt the model to start reflecting once they appear; increasing or decreasing the frequency of these keywords effectively increases or decreases the frequency of reflection. [1] shows that simply inserting reflective keywords can elicit reflection even in pre-trained models, likely because they frequently co-occur with reflective reasoning in the pre-training corpus. [2] further finds that the hidden states preceding reflective keywords differ systematically from other hidden states, and that linearly intervening along the difference between these states can encourage or suppress reflection. Building on this observation, several decoding methods explicitly modify the probabilities of reflection to control how often models reflect [3-6], and their successes provide further evidence for the strong link between reflective tokens and reflection frequency. Together, they show that these keywords are tightly coupled with reflection, because they are highly correlated with reflective behavior in the pre-training data, giving them great robustness as trigger words to influence genuine self-correction across diverse models.
>
> The above discussion finds that the frequency of reflective keywords is highly correlated with the model's actual reflection frequency, ensuring that, in most cases, our reward can function as intended. However, online RL can indeed be more fragile. In rare cases where a trajectory contains many reflective keywords but still exhibits little genuine reflection, the model might increase the probability of that trajectory and potentially amplify such special cases in the future. To mitigate this, we apply three concrete design choices:

---

> ### Author Response · Authors · 2025-11-27
> **Official Comment to Reviewer homH (3/4)**
>
> 1. **Span-based counting instead of raw keyword counts.**
>  We do not simply count the raw occurrences of reflective keywords. Instead, we treat consecutive occurrences within a window of 60 characters as a single instance. The goal is to approximate the number of reflective sentences rather than a token-level count. This makes the reward more robust and avoids pathologies such as “Wait, wait, but wait,” receiving a high reflection score.
>
> 2. **Penalizing only clearly insufficient reflection.**
>  We do not always encourage more reflective keywords. We only penalize trajectories when reflection is clearly insufficient—i.e., when the keyword frequency falls below 80% of the average level， in order to avoid completely skipping reflection due to the lack of trigger words. In other cases, where there is already sufficient reflection, we do not reward high reflection frequency. As a result, only a very small fraction of trajectories with clearly insufficient reflection are penalized, and we do not encourage arbitrarily high keyword frequencies, preventing the keyword frequency from continuously increasing.
>
> 3. **Regularization to prevent drift.**
>  We apply a KL-divergence constraint and use LoRA to prevent the learned policy from drifting too far from the base model, which further reduces the risk of such issues.
>
> Therefore, for training data where reflective keywords are rare but genuine reflection is still rich, our reward may occasionally assign an imperfect penalty, but this does not lead to reward hacking, and the 80% threshold minimizes such cases. When reflective keywords are frequent but genuine reflection is limited, if this leads to an incorrect answer, both the accuracy reward and the length reward become zero, so such trajectories are effectively penalized. If the answer is correct, we view this as acceptable and do not penalize them. Overall, our reward design ensures that incorrect trajectories are penalized in all cases, thereby avoiding reward hacking. In our extensive experiments, we did not observe any reward hacking phenomena.
>
> The more semantic metric you suggest would likely work as well, but it would introduce additional computational overhead. Our approach achieves sufficiently good performance without extra latency, so we chose the current design. Thank you again for this insightful suggestion.
>
> [1] Rethinking Reflection in Pre-Training, arXiv 2025. 35 citations.
>
> [2] From Emergence to Control: Probing and Modulating Self-Reflection in Language Models, arXiv 2025.
>
> [3] Efficient Reasoning for Large Reasoning Language Models via Certainty-Guided Reflection Suppression, AAAI 2026.
>
> [4] Speculative Thinking: Enhancing Small-Model Reasoning with Large Model Guidance at Inference Time, COLM 2025.
>
> [5] Thoughts Are All Over the Place: On the Underthinking of o1-Like LLMs, arXiv 2025. 83 citations.
>
> [6] Missing Premise Exacerbates Overthinking: Are Reasoning Models Losing Critical Thinking Skill? COLM 2025.
>
> > Q1: I’m curious about the training setup: the paper mentions that both the original and revised responses are used together during online updates. Have the authors tried using only the revised responses for training? It would be interesting to see whether excluding the original (possibly overthinking) trajectories leads to more stable learning or better efficiency–accuracy trade-offs.
>
> We sincerely thank you for this suggestion — this setting gave us additional insight. We tried two different configurations:
>
> 1. directly applying supervised fine-tuning on the non-overthinking trajectories, and
> 2. starting from the GRPO ($R_{RLen+Reflect}$ $M_{Reflect}$) setup and removing all original trajectories (Revision Path Only).

---

> ### Author Response · Authors · 2025-11-27
> **Official Comment to Reviewer homH (4/4)**
>
> In each batch, we use 16 training examples and sample 4 trajectories per example. We train for 1,800 steps, and all other hyperparameters are kept exactly the same. The results are as follows.
>
> | | GSM8K Acc↑ |  GSM8K TR↓ | Math500 Acc↑ | Math500 TR↓ | Gaokao23 Acc↑ | Gaokao23 TR↓ | Amc23 Acc↑ |  Amc23 TR↓ | Aime24 Acc↑ | Aime24 TR↓ |  Avg Acc↑ | Avg TR↓ |
> | -- | -: | -: | --: | -: | -: | --: | -: | -: | -: | -: | --: | -: |
> | **Budget: 8k** |
> | Original | 91.58 | 100$_{1312}$ |  88.40 |  100$_{3389}$ | 77.40 | 100$_{3270}$ | 77.50 | 100$_{4746}$ | 40.83 | 100$_{6815}$ |  75.14 | 100$_{3906}$ |
> | SFT  | 89.92 | 64.63 |  89.20 | 69.25 | 77.92 |  72.91 | 82.81 | 72.67 | 41.25 | 89.98 |  76.22 | 73.89 |
> | RPO  | 90.14 | 51.22 |  89.80 | 53.35 | 83.12 |  55.20 | 80.62 | 59.54 | 41.67 | 74.78 |  77.07 | 58.82 |
> | GRPO $R_{Len}$ | 85.97 | 32.01 |  87.60 | 59.04 | 70.91 |  58.84 | 85.62 | 71.43 | 45.42 | 87.42 |  75.10 | 61.75 |
> | GRPO $R_{Len}$ + $M_{Reflect}$  | 89.23 | 37.80 |  91.20 | 48.66 | 78.44 |  52.23 | 86.56 | 59.65 | 41.67 | 84.15 |  77.42 |  56.50 |
> | GRPO $R_{RLen+Reflect}$ | 92.72 | 67.07 |  91.40 | 69.52 | 80.52 |  72.54 | 85.62 | 75.64 | 47.92 | 91.37 | 79.64 | 75.23 |
> | GRPO $R_{RLen+Reflect}$ + $M_{Reflect}$  | 89.99 | 53.12 |  89.60 | 61.79 | 81.30 |  61.19 | 85.62 | 71.39 | 42.92 | 88.45 | 77.89 | 67.19 |
> | Revision Path Only | 89.84 | 44.82 |  91.20 | 55.83 | 76.62 |  59.11 | 81.56 | 62.68 | 44.17 | 84.48 |  76.68 | 61.38 |
> |**Budget: 16k**|
> | Original | 91.66 | 100$_{1344}$ |  92.00 |  100$_{3893}$ | 81.82 | 100$_{3785}$ | 88.12 | 100$_{5840}$ | 48.33 | 100$_{10460}$ |  80.39 | 100$_{5064}$ |
> | SFT | 89.99 | 68.75 |  90.60 | 69.25 | 79.48 |  74.37 | 88.44 | 72.11 | 48.75 | 89.68 |  79.45 | 74.83 |
> | RPO | 90.14 | 50.00 |  89.80 | 46.90 | 83.12 |  47.77 | 82.19 | 50.75 | 42.92 | 51.41 |  77.63 | 49.37 |
> | GRPO $R_{Len}$ | 85.97 | 31.25 |  88.40 | 56.43 | 72.21 |  55.88 | 87.81 | 65.65 | 50.00 | 76.96 |  76.88 | 57.23 |
> | GRPO $R_{Len}$ + $M_{Reflect}$ | 89.23 | 36.90 |  92.40 | 45.11 | 79.74 |  47.50 | 88.12 | 56.04 | 47.92 | 75.82 |  79.48 | 52.27 |
> | GRPO $R_{RLen+Reflect}$ | 92.72 | 65.48 |  92.80 | 66.71 | 81.82 |  68.27 | 88.75 | 72.17 | 54.58 | 86.21 | 82.13 | 71.77 |
> | GRPO $R_{RLen+Reflect}$ + $M_{Reflect}$ | 89.99 | 52.08 |  91.00 | 60.78 | 82.08 |  55.85 | 89.38 | 67.76 | 51.25 | 81.09 |  80.74 |  63.51 |
> | Revision Path Only | 89.92 | 43.90 |  92.20 | 53.63 | 78.70 |  58.68 | 87.50 | 61.23 | 52.08 | 77.20 |  80.08 | 58.93 |
>
> The SFT setting performs poorly. This is because the SFT training paradigm cannot provide online feedback: the model mainly learns a superficial pattern of “being able to stop at any time,” instead of learning how to simultaneously keep outputs both short and correct, or how to further improve an already shortened output.
>
> However, in the GRPO setting, removing the original trajectories has only a minor impact on performance: both the length and the token ratio decrease slightly, which does not clearly establish a strong advantage for either including or excluding them. At the same time, when we remove the trajectories generated by the reflection model (the “$R_{RLen+Reflect}$” variant), the performance is also quite strong, and both the length and token ratio increase slightly.
>
> This shows that using only the parallel-sampling trajectories or only the sequential-revision trajectories with the reflection model can both yield strong results. The difference is that sequential revision tends to encourage shorter generations, whereas parallel sampling offers limited gains in efficiency. There are two main reasons for this:
>
> 1. After pruning, the training data contain a style of shorter trajectories; encouraging such trajectories helps the model shorten its outputs more quickly.
> 2. After removing the parallel-sampling paths, the length reward is calculated only on the *effective* part of the trajectory, rather than on the entire trajectory. This encourages solutions that obtain the answer quickly. In the basic setup, a trajectory might reach the answer quickly but then continue with a long reflection segment, which is penalized by the length reward.
>
> Therefore, when we train using the sequential-revision trajectories, the resulting paths are relatively shorter. However, the different configurations do not exhibit large performance gaps; they primarily reflect different trade-offs between accuracy and efficiency. Thanks again for your insightful suggention.

---

### Official Review · Reviewer_mhkM · 2025-11-01

**Soundness:** 2
**Presentation:** 3
**Contribution:** 2
**Rating:** 4
**Confidence:** 3

**Summary:**

The paper introduces REA-RL, a reflection-aware online reinforcement learning framework that improves the efficiency–accuracy trade-off of large reasoning models by curbing overthinking without discarding beneficial reflection. The core idea is to augment standard grouped online RL with two complementary mechanisms. First, a lightweight reflection model is trained to detect the first point in a sampled trajectory where the answer is already present; tokens after that point are treated as overthinking and the path is revised by truncating the think segment and prompting the policy to finalize the answer. This yields a sequential-revision signal that, when combined with parallel sampling, creates an additional scaling dimension and can be interpreted as a partial advantage that penalizes overthinking tokens while rewarding the revised completions. Second, a reflection-aware reward measures the density of reflective cues (e.g., wait, but) and penalizes paths whose reflection density falls in the lowest quantiles, thereby preventing the length reward from collapsing the model into non-reflective, error-prone plans. The authors also refine the length reward by zeroing it on incorrect generations. Across five math benchmarks and two generation budgets, the approach maintains or improves accuracy while reducing token cost by up to 36% on average relative to the original R1-7B initialization, and analysis shows it preserves reflection on harder problems while appropriately reducing it on easier ones. A workflow schematic on page 5 clarifies how parallel sampling, reflection-guided truncation, and reward shaping interact during training, and tables on pages 6–8 report the main and component-wise results supporting these claims.

**Strengths:**

The work is original in how it operationalizes reflection within online RL: rather than only rewarding short outputs, it explicitly detects and trims overthinking in-situ and then optimizes on both original and revised trajectories. This integration of parallel sampling with sequential revision is conceptually clean, computationally practical, and linked to an interpretable partial-advantage view that clarifies why overthinking tokens receive targeted penalties while preserving valid reasoning.

Method quality is supported by a carefully designed reflection model distilled to 7B for speed, a simple yet effective reflection-density reward, and a refined length reward that avoids incentivizing short but wrong responses. The empirical study is broad for the domain, covering GSM8K, MATH500, Gaokao23, AMC’23, and AIME’24 with two budgets, with ablations isolating the effects of reflection modeling versus reward shaping and training-dynamics plots that explain how the method shortens traces without eroding accuracy.

Clarity is aided by step-by-step descriptions, concrete prompts, and a readable workflow diagram on page 5; the case studies on page 16 make the truncation–revision mechanism tangible.

**Weaknesses:**

The reliance on answer-presence detection as the stopping criterion risks truncating useful verification steps when the answer is mentioned early in a speculative way, and while the authors mitigate this with a trained reflection model, there is limited quantitative reporting on its detection precision/recall beyond downstream accuracy and token ratios.

The study focuses on math word problems with a single distilled 7B base; it remains unclear whether the approach scales to other reasoning domains, like multimodal or non-math reasoning, or to larger pretrained LRMs without distillation artifacts, especially with a reflection model trained for the specific in-distribution task.

The online pipeline introduces extra complexity and additional training time relative to pure parallel sampling, and although the gains seem to justify this, a more detailed wall-clock and latency analysis at inference would strengthen the case.

 Some baselines that aggressively shorten chains show larger efficiency but lower accuracy; while the paper explains the trade-offs, a controlled budget-equalized comparison could more cleanly quantify accuracy at equal compute.

**Questions:**

See weaknesses

---

> ### Author Response · Authors · 2025-11-27
> **Official Comment to Reviewer mhkM (1/4)**
>
> > W1: The reliance on answer-presence detection as the stopping criterion risks truncating useful verification steps when the answer is mentioned early in a speculative way, and while the authors mitigate this with a trained reflection model, there is limited quantitative reporting on its detection precision/recall beyond downstream accuracy and token ratios.
>
> Thank you for the suggestion. In fact, we have already included these experiments, with details in **Table 3**. Since answer-presence detection lacks access to the gold answer, we cannot directly measure its precision. Instead, after truncation, we force the model to **generate the final answer**, and then analyze the change in accuracy and output length before and after truncation, as described in Section 3.2.
>
> We copy the average performance below. We present results for the 32B and 7B models using the two-stage detection method in Section 3.1 without training (7B / 32B Revise), and for our trained 7B reflection model ($M_{Reflect}$). The results demonstrate that 32B Revise can reduce model output length with almost no loss in capability, and that the $M_{Reflect}$ achieves better results than the 7B Revise without training and the fixed truncation strategy. Further, its results are close to those of the teacher model (32B) but retain more tokens, representing slightly worse but comparable truncation performance. These results show that our method can effectively reduce the number of output tokens while maintaining accuracy, which demonstrates the effectiveness of our approach.
>
> | Method  | Avg Acc↑  | Avg TR↓  |
> |-------------------------------------------------------------|-----------------|------------------|
> | Original  | 80.39 | 100$_{5064}$ |
> | 7B Revise | 75.77 | 69.65  |
> | + Gold  | 74.02 | 61.15  |
> | 32B Revise  | 80.61 | 75.65  |
> | + Gold  | 79.82 | 66.32  |
> | $M_{Reflect}$ | 80.14 | 78.52  |
> | Fixed Trunc (78.93) | 76.86 | 79.68  |
> | GRPO $R_{RLen+Reflect}$ $M_{Reflect}$ | 80.74 | 63.51 |
>
> > W2: The study focuses on math word problems with a single distilled 7B base; it remains unclear whether the approach scales to other reasoning domains, like multimodal or non-math reasoning, or to larger pretrained LRMs without distillation artifacts, especially with a reflection model trained for the specific in-distribution task.
>
> Thank you for your suggestions, which help us improve our paper. Our method is able to transfer to different reasoning domains as well as different model families and scales. To demonstrate the improvement of our approach on a broader range of **reasoning domains** and **additional models**, we extend our experiments to general QA reasoning and different policy models. We also train a **reflection model for the specific general QA domains**.
>
> **Models.** Because GRPO training is time- and resource-intensive, we intentionally use smaller models that differ substantially from the main model to verify generality. Specifically, we add experiments on **SmolLM3-3B** [1], a non-DeepSeek, non-Qwen model series pre-trained from scratch **without distillation artifacts**. It was released by Hugging Face on September 10 and has received 69k downloads. We also include **DeepSeek-R1-Distill-Qwen-1.5B (Deepseek-1.5B)** to further validate the effectiveness of our method across different scales.
>
> **Datasets.** We add experiments on general QA reasoning. We follow [2, 3] to evaluate on **MMLU** [4], **MMLU-Pro** [5], and **SuperGPQA** [6], covering different difficulty levels. Based on the size of their original datasets, we select 20k examples from MMLU, 5k from MMLU-Pro, and 15k from SuperGPQA to construct a 40k mixed training set and 5k examples from each dataset as test sets. Since some QA questions do not require deep reasoning, we first filter out the 15% of questions with the shortest paths as generated by DeepSeek-1.5B, and then sample the required number of examples. Finally, for the training set, we remove questions that DeepSeek-1.5B cannot solve within 4 paths, resulting in a training set of 30k examples.
>
> **Reflection model for general QA.** We train a new reflection model to show that it can generalize to **other reasoning domains** and **additional model scales**. Concretely, we generate SFT data using a similar procedure as in the main text, with two differences: (1) we use DeepSeek-1.5B to generate the original trajectories, and (2) we use Qwen3-32B as the reviser. These data are then mixed with the math-domain data generated in the main text and used together as the training set. We subsequently train reflection models of different sizes, based on **Qwen2.5-7B-Instruct** (as in the main text) and **Qwen3-4B**, to show that our method also works well with smaller base models.

---

> ### Author Response · Authors · 2025-11-27
> **Official Comment to Reviewer mhkM (2/4)**
>
> **Reflection model evaluation.** After training, we evaluate using the same protocol as in Section 3.2: we apply the trained reflection models to truncate the trajectories and then force the model to **generate the final answer**. We test the effectiveness of the truncation model by comparing the accuracy and path length before and after truncation. All tests below are conducted on **MMLU-Pro**, which has medium difficulty. The results are in the table below. Each column corresponds to a different generation model being evaluated, and each row corresponds to a different reflection model. *Base* denotes the performance before truncation. *Normal* truncates at the first identified correct answer; *Weak* truncates at the second such position; and *Strong* truncates before the first position where the truncation probability exceeds 0.25. Acc denotes accuracy, and TR denotes the token ratio (a lower TR indicates higher efficiency).
>
> |  | DeepSeek-1.5B Acc↑ | DeepSeek-1.5B TR↓ | DeepSeek-7B Acc↑ | DeepSeek-7B TR↓ | Avg Acc↑ | Avg TR↓ |
> | ---- | --- | --- | --- | ---- | - | -- |
> | Base | 30.30 | 100.00$_{4844}$ | 49.32 | 100.00$_{3834}$ | 39.81 | 100.00$_{4339}$ |
> | Qwen2.5-7B-Instruct-Weak | 30.62 | 100.72 | 50.42 | 99.40  | 40.52 | 100.06 |
> | Qwen3-4B-Weak  | 30.48 | 101.96 | 51.60 | 99.48  | 41.04 | 100.72 |
> | Qwen2.5-7B-Instruct-Normal | 31.40 | 95.11  | 50.12 | 92.96  | 40.76 | 94.04  |
> | Qwen3-4B-Normal  | 31.40 | 96.26  | 50.42 | 94.31  | 40.91 | 95.29  |
> | Qwen2.5-7B-Instruct-Strong | 30.60 | 89.91  | 48.44 | 87.14  | 39.52 | 88.53  |
> | Qwen3-4B-Strong  | 31.16 | 89.88  | 48.70 | 87.48  | 39.93 | 88.68  |
>
> We observe that, whether we use a **7B** or **4B** reflection model, the performance is almost identical. Under the Normal setting, all reflection models can effectively truncate reasoning paths in the general QA domain, demonstrating that our method is robust to the reasoning domains and the base reflection model. Smaller reflection models can still achieve strong results.
>
> **Training.** We use the same training configuration as in the main text to compare the effect of using only the length reward versus combining it with the reflection model. The prompt still uses the same setting as in the main text, i.e., “Please reason step by step, and put your final answer within \boxed{}. Question: {Question}.” We only replace the question with the concatenation of the question and choices. After training, since the dataset is large enough, we generate one path for each question and report the average results, with generation hyperparameters consistent with the main text. In addition, in SmolLM3-3B, repeated generations occasionally occur, so we add a frequency_penalty of 0.05 to alleviate this. The results demonstrate that introducing the reflection model brings stable improvements, enhancing both accuracy and efficiency.
>
> | | MMLU Acc↑ | MMLU TR↓ | MMLU Pro Acc↑ | MMLU Pro TR↓ | SuperGPQA Acc↑ | SuperGPQA TR↓ | Avg Acc↑ | Avg TR↓ |
> | --- | -- | - | -- | -- | -- | -- | - | -- |
> | **Budget: 8k** |
> | DeepSeek-1.5B  | 40.42  | 100$_{1568}$ | 27.44  | 100$_{3632}$ | 13.52 | 100$_{4441}$  | 27.13 | 100$_{3214}$  |
> | GRPO $R_{Len}$ | 40.36  | 74.55 | 28.10  | 85.77 | 11.98 | 98.13  | 26.81 | 86.15  |
> | GRPO $R_{Len}$ + $M_{Reflect}$ | 40.94  | 73.85 | 27.88  | 89.98 | 13.56 | 93.22  | 27.46 | 85.68  |
> | SmolLM3-3B | 52.44  | 100$_{2615}$ | 34.84  | 100$_{3966}$ | 17.94 | 100$_{4369}$  | 35.07 | 100$_{3650}$  |
> | GRPO $R_{Len}$ | 52.02  | 73.61 | 34.02  | 83.59 | 16.76 | 87.48  | 34.27 | 81.56  |
> | GRPO $R_{Len}$ + $M_{Reflect}$ | 54.79  | 65.20 | 34.90  | 80.81 | 16.34 | 84.96  | 35.34 | 76.99  |
> | **Budget: 16k**  |
> | DeepSeek-1.5B  | 40.98  | 100$_{1871}$ | 30.30  | 100$_{4844}$ | 16.66 | 100$_{6337}$  | 29.31 | 100$_{4351}$  |
> | GRPO $R_{Len}$ | 40.58  | 70.82 | 30.24  | 84.56 | 15.36 | 104.92 | 28.73 | 86.77  |
> | GRPO $R_{Len}$ + $M_{Reflect}$ | 41.30  | 74.77 | 30.48  | 89.66 | 16.86 | 91.68  | 29.55 | 85.37  |
> | SmolLM3-3B | 53.54  | 100$_{2985}$ | 36.92  | 100$_{4857}$ | 19.34 | 100$_{5398}$  | 36.60 | 100$_{4413}$  |
> | GRPO $R_{Len}$ | 53.14  | 72.63 | 35.88  | 86.43 | 18.56 | 92.92  | 35.86 | 83.99  |
> | GRPO $R_{Len}$ + $M_{Reflect}$ | 55.77  | 62.04 | 36.60  | 79.16 | 18.14 | 84.94  | 36.84 | 75.38  |
>
> These experiments demonstrate that our method performs well across different domains (math and general QA), different model families, and different scales (1.5B–7B). We hope these additional results alleviate the generalization concerns.
>
> [1] SmolLM3: smol, multilingual, long-context reasoner. Hugging Face 2025.
>
> [2] Qwen3 Technical Report. arXiv 2025.
>
> [3] R-Zero: Self-Evolving Reasoning LLM from Zero Data. arXiv 2025. 24 citations.
>
> [4] Measuring Massive Multitask Language Understanding. ICLR 2021.
>
> [5] MMLU-Pro: A More Robust and Challenging Multi-Task Language Understanding Benchmark. NeurIPS (Datasets & Benchmarks) 2024.
>
> [6] SuperGPQA: Scaling LLM Evaluation across 285 Graduate Disciplines. arXiv 2025.

---

> ### Author Response · Authors · 2025-11-27
> **Official Comment to Reviewer mhkM (3/4)**
>
> > W3: The online pipeline introduces extra complexity and additional training time relative to pure parallel sampling, and although the gains seem to justify this, a more detailed wall-clock and latency analysis at inference would strengthen the case.
>
> Thank you for the suggestion; this helps us better characterize the cost of our approach. During training, our extra wall-clock time cost comes from **sequential revision**. Although our method requires more training time, it actually consumes less wall-clock time for sequential revision than parallel sampling and has the potential to improve efficiency when deployed at scale. The analysis is detailed as follows.
>
> **Wall-clock training time.** In the main text, we already provide a comprehensive efficiency-related discussion in Section 5.2. Specifically, on 3 NVIDIA A800 80G GPUs, GRPO with sampling 4 paths requires 80 hours, while sampling 8 paths requires 110 hours. When incorporating the reflection model, which samples 4 paths and then revises them to 8 paths, it requires 120 hours. After introducing the reflection model, we obtain, on average, a 3% performance improvement and 4% shorter outputs compared to sampling 8 paths. Given the improvement, this additional cost is acceptable. If training efficiency is a primary concern, utilizing the reflection reward alone also requires only 80 hours and still achieves improved performance.
>
> **Token cost analysis.** To provide a more thorough analysis of the latency introduced by the reflection model, we analyze token consumption and rollout latency. For parallel sampling, the cost of the rollout is to input the question and output a complete reasoning path. For sequential revision, we instead input the full path and output a label for each path from the reflection model, and then input the truncated path and output a very short suffix as the final answer from the policy model.
>
> We measure the number of tokens used to generate a single path under the two schemes during training. For parallel sampling, the average input length is 94 tokens, and the average output length is 3605 tokens. For sequential revision, the input is 6678 tokens and the output is 319 tokens. Given that our method significantly reduces the output length, this substantially reduces the cost of inference, because input tokens are typically at most 1/5 the price of output tokens. For example, for gemini-3-pro, the input price is \\$2.00 per 1M tokens (\\$0.20 for cached input), while the output price is \\$12.00 per 1M tokens; for gpt-5, the input price is \\$1.25 per 1M tokens (\\$0.125 for cached input), and the output price is \\$10.00 per 1M tokens. In particular, about half of our input tokens to the policy model are cached during parallel sampling (suffix generation by the policy model), which makes it even cheaper. Concretely, using the GPT-5 pricing as an example, parallel sampling for 1k examples costs \\$36.17, whereas sequential revision for 1k examples costs \\$7.78. This shows that, under an ideal large-scale generation scenario, the cost of our method is lower.
>
> **Latency analysis.** We also measure small-scale vLLM inference time on a **per-batch** basis, which aligns with our training setup. On a single A800 GPU, generating one batch of data with 4 paths takes 158s. Our method takes 47s to generate labels plus 2s to generate answers, for a total of 49s for sequential revision, which is less than one third of the traditional rollout. This demonstrates that the cost of sequential revision is clearly lower than that of standard parallel sampling.
>
> Therefore, although we spend slightly more total **training** time (110h vs 120h), we actually reduce the generation cost by converting expensive output tokens into cheaper input tokens. However, for online generation per batch, the three generation stages in our method must run sequentially, where each batch spends 207s  (158 + 47 + 2) on inference on average. In contrast, standard generation of 4 paths takes 158s on average, and due to better parallelism, generating 8 paths takes 192s on average, which is shorter than our 207s and therefore leads to a shorter overall training time (110h vs 120h). Nevertheless, we believe that under large-scale implementations (e.g., dedicated data-generation and training servers [1-3]), the training time of our method can be lower.
>
> **Test time inference.** At test time, our trained model does not use the reflection model. This is because using the reflection model requires that a complete reasoning path has already been generated, which makes it inherently less efficient than direct generation. Therefore, the fact that our method consumes fewer tokens directly implies that it requires less wall-clock time and lower latency during test time. As shown in the table below, we summarize the wall-clock minutes spent by vLLM for full generation on the test sets. The time savings closely track the reduction in tokens, which supports our claim.

---

> ### Author Response · Authors · 2025-11-27
> **Official Comment to Reviewer mhkM (4/4)**
>
> | Method  | GSM8K | Math500 | Gaokao23 | Amc23 | Aime24 | Avg |
> | -- | -: | -: | -: | -: | --: | -: |
> | Base (minute) |  18.7 | 36.3 |  27.5 |  41.5 | 73.0 |  39.4 |
> | Ours (minute) |  10.7 | 23.6 |  17.8 |  28.1 | 61.7 |  28.4 |
> | Time Reduction (%)  | 42.78 | 34.99 | 35.27 | 32.29 |  15.48 | 27.92 |
> | Token Reduction (%) | 47.92 | 39.22 | 44.15 | 32.24 |  18.91 | 36.49 |
>
> [1] AReaL: A Large-Scale Asynchronous Reinforcement Learning System for Language Reasoning. arXiv 2025. 37 citations.
>
> [2] LlamaRL: A Distributed Asynchronous Reinforcement Learning Framework for Efficient Large-scale LLM Training. arXiv 2025. 9 citations.
>
> [3] Echo: Decoupling Inference and Training for Large-Scale RL Alignment on Heterogeneous Swarms. arXiv 2025.
>
> > W4: Some baselines that aggressively shorten chains show larger efficiency but lower accuracy; while the paper explains the trade-offs, a controlled budget-equalized comparison could more cleanly quantify accuracy at equal compute.
>
> Thank you for pointing this out; this helps us provide clearer analysis. Accuracy and length are inherently in a trade-off relationship, and controlling one dimension to compare the other can more clearly show the advantages of our method. However, this is difficult to control in practice. It is hard to make models learned under different methods exactly match on **either** accuracy or output length.
>
> One possible method would be to truncate the reasoning at a fixed length and then force the model to generate the final answer to measure accuracy. However, this goes against the main motivation of our paper. We aim for the model to **know when to stop**, rather than to give the correct answer along the *shortest* path under an externally imposed cutoff. If we manually truncate sequences, we can no longer test whether the model itself knows when to stop, and thus, such results would provide limited evidence for our main contribution.
>
> To address this, we analyze the problem along two dimensions. First, compared with the following three baselines, at least one of our methods achieves **better performance and higher efficiency** at the same time, which supports the effectiveness of our approach:
>
> | Method                                | Avg Acc↑ | Avg TR↓ |
> | ------------------------------------- | -------: | ------: |
> |**Budget: 8k**|
> | Light-R1 |    71.91 |   78.29 |
> | DAST |    76.97 |   74.96 |
> | Arora et al. |    76.72 |   72.62 |
> | GRPO $R_{RLen+Reflect}$ $M_{Reflect}$ |  77.89 |   67.19 |
> |**Budget: 16k**|
> | Light-R1 |    79.07 |   76.50 |
> | DAST |    80.48 |   76.17 |
> | Arora et al. |    79.20 |   70.71 |
> | GRPO $R_{RLen+Reflect}$ $M_{Reflect}$ |    80.74 |   63.51 |
>
> For two other baselines that aggressively shorten chains, the accuracy drops are too large, especially when the generation budget is sufficient (8–15 points compared to the original model on average under 16k budget), which we consider unacceptable. Compared with these baselines, **GRPO $R_{Len}$** achieves a smaller performance drop (-3.5 under 16k budget) and higher efficiency than two of our methods (43% improvement on efficiency). Therefore, we use GRPO $R_{Len}$ as a representative for the high efficiency baseline to perform a more controlled comparison against our method.
>
> Since GRPO $R_{Len}$ is trained by us, we can choose checkpoints to match on one metric and compare the other. As shown in Figure 3, the two best checkpoints for GRPO $R_{Len}$ are:
>
> * **Step 1000**: accuracy **80.12** & token ratio **68.96**
> * **Step 1200**: accuracy **78.74** & token ratio **63.96**
>
> Before step 1000, the accuracy is close to that at step 1000, but the inference efficiency (TR) is much worse (higher TR). After step 1200, the accuracy drops significantly while the efficiency does not improve much. Based on this, we select comparable checkpoints for our method GRPO $R_{RLen+Reflect}$ $M_{Reflect}$, which has higher accuracy but lower efficiency:
>
> * For the step-1000 GRPO $R_{Len}$ checkpoint (80.12 & 68.96 in terms of accuracy & TR), our method at step 1600 achieves similar efficiency (81.17 & 69.39) but **higher accuracy**. At step 1800, our method achieves similar accuracy (80.74 & 63.51 for accuracy & TR) but **better efficiency**.
> * For the step-1200 GRPO $R_{Len}$ checkpoint (78.74 & 63.96), our method at step 1800 again reaches similar efficiency (80.74 & 63.51) but **higher accuracy**.
>
> This controlled comparison demonstrates the advantages of our approach under a roughly equal compute budget. Our method using only the reflection model, GRPO $R_{Len}$ $M_{Reflect}$, achieves both higher accuracy and higher efficiency compared to length reward only. Hope this alleviates your concern.

---

### Official Review · Reviewer_xPcf · 2025-11-03

**Soundness:** 2
**Presentation:** 3
**Contribution:** 2
**Rating:** 4
**Confidence:** 4

**Summary:**

This paper proposes **REA-RL (Reflection-Aware Reinforcement Learning)**, a framework for improving reasoning efficiency in Large Reasoning Models (LRMs) without sacrificing accuracy. The authors identify the problem of *overthinking*—excessive reflection that increases inference cost—and introduce two key components: a **reflection model** to detect and remove redundant reasoning in real-time, and a **reflection reward** to preserve necessary reflective behavior. Combined, these techniques reduce inference token usage by **36%** while maintaining or improving performance across several reasoning benchmarks, offering a balanced solution between efficiency and reflection quality.

**Strengths:**

1. The paper targets an important and practical issue—over-reflection in large reasoning models.
2. The proposed REA-RL framework is conceptually clear.
3. The writing is easy to follow.
4. The experiments demonstrate the effectiveness of the proposed methods.

**Weaknesses:**

1. The paper evaluates only on the MATH domain, and does not include experiments in other reasoning domains (e.g., code, general QA, agentic tasks), which limits the demonstrated generality of the approach.
2. All experiments are conducted solely on R1-Qwen-7B; including additional model families and scales would strengthen the empirical evidence and show broader applicability.
3. The method introduces extra computational overhead, requiring double rollouts and an additional reflection-model inference.

**Questions:**

1. This work shows that the reflection model $M_{reflect}$ contributes to the efficiency–efficacy trade-off. Would it be possible to include a baseline that simply truncates the reasoning sequence when the first reflection token appears and then appends </think> to complete the response from that point?
2. Compared with methods using only length-based rewards, approaches involving \( M_{\text{reflect}} \) seem to yield slightly lower accuracy. Do the authors have any insights into why this happens?
3. In line 319, the paper claims that GRPO with accuracy-only rewards does not further improve accuracy, which seems inconsistent with previous findings such as *DeepScaleR* [1]. Could the authors clarify this discrepancy or provide an explanation?

[1] *DeepScaleR: Surpassing O1-Preview with a 1.5B Model by Scaling RL.* https://pretty-radio-b75.notion.site/DeepScaleR-Surpassing-O1-Preview-with-a-1-5B-Model-by-Scaling-RL-19681902c1468005bed8ca303013a4e2

---

> ### Author Response · Authors · 2025-11-27
> **Official Comment to Reviewer xPcf (1/5)**
>
> > W1: The paper evaluates only on the MATH domain, and does not include experiments in other reasoning domains (e.g., code, general QA, agentic tasks), which limits the demonstrated generality of the approach.
>
> > W2: All experiments are conducted solely on R1-Qwen-7B; including additional model families and scales would strengthen the empirical evidence and show broader applicability.
>
> Thank you for your suggestions, which help us improve our paper. Our method is able to transfer to different reasoning domains as well as different model families and scales. To demonstrate the improvement of our approach on a broader range of **reasoning domains** and **additional model families and scales**, we extend our experiments to general QA reasoning and different policy models.
>
> **Models.** Because GRPO training is time- and resource-intensive, we intentionally use smaller models that differ substantially from the main model to verify generality. Specifically, we add experiments on **SmolLM3-3B** [1], a non-DeepSeek, non-Qwen model series pre-trained from scratch. It was released by Hugging Face on September 10 and has received 69k downloads. We also include **DeepSeek-R1-Distill-Qwen-1.5B (Deepseek-1.5B)** to further validate the effectiveness of our method across different scales.
>
> **Datasets.** We add experiments on general QA reasoning. We follow [2, 3] to evaluate on **MMLU** [4], **MMLU-Pro** [5], and **SuperGPQA** [6], covering different difficulty levels. Based on the size of their original datasets, we select 20k examples from MMLU, 5k from MMLU-Pro, and 15k from SuperGPQA to construct a 40k mixed training set and 5k examples from each dataset as test sets. Since some QA questions do not require deep reasoning, we first filter out the 15% of questions with the shortest paths as generated by DeepSeek-1.5B, and then sample the required number of examples. Finally, for the training set, we remove questions that DeepSeek-1.5B cannot solve within 4 paths, resulting in a training set of 30k examples.
>
> **Reflection model for general QA.** We train a new reflection model to show that the reflection model can generalize to **other reasoning domains** and **additional model scales**. Concretely, we generate SFT data using a similar procedure as in the main text, with two differences: (1) we use DeepSeek-1.5B to generate the original trajectories, and (2) we use Qwen3-32B as the reviser. These data are then mixed with the math-domain data generated in the main text and used together as the training set. We subsequently train reflection models of different sizes, based on **Qwen2.5-7B-Instruct** (as in the main text) and **Qwen3-4B**, to show that our method also works well with smaller base models.
>
> **Reflection model evaluation.** After training, we evaluate using the same protocol as in Section 3.2: we apply the trained reflection models to truncate the trajectories and then force the model to **generate the final answer**. We test the effectiveness of the truncation model by comparing the accuracy and path length before and after truncation. All tests below are conducted on **MMLU-Pro**, which has medium difficulty. The results are in the table below. Each column corresponds to a different generation model being evaluated, and each row corresponds to a different reflection model. *Base* denotes the performance before truncation. *Normal* truncates at the first identified correct answer; *Weak* truncates at the second such position; and *Strong* truncates before the first position where the truncation probability exceeds 0.25. Acc denotes accuracy, and TR denotes the token ratio (a lower TR indicates higher efficiency).
>
> |  | DeepSeek-1.5B Acc↑ | DeepSeek-1.5B TR↓ | DeepSeek-7B Acc↑ | DeepSeek-7B TR↓ | Avg Acc↑ | Avg TR↓ |
> | ---- | --- | --- | --- | ---- | - | -- |
> | Original | 30.30 | 100.00$_{4844}$ | 49.32 | 100.00$_{3834}$ | 39.81 | 100.00$_{4339}$ |
> | Qwen2.5-7B-Instruct-Weak | 30.62 | 100.72 | 50.42 | 99.40  | 40.52 | 100.06 |
> | Qwen3-4B-Weak  | 30.48 | 101.96 | 51.60 | 99.48  | 41.04 | 100.72 |
> | Qwen2.5-7B-Instruct-Normal | 31.40 | 95.11  | 50.12 | 92.96  | 40.76 | 94.04  |
> | Qwen3-4B-Normal  | 31.40 | 96.26  | 50.42 | 94.31  | 40.91 | 95.29  |
> | Qwen2.5-7B-Instruct-Strong | 30.60 | 89.91  | 48.44 | 87.14  | 39.52 | 88.53  |
> | Qwen3-4B-Strong  | 31.16 | 89.88  | 48.70 | 87.48  | 39.93 | 88.68  |
>
> We observe that, whether we use a **7B** or **4B** reflection model, the performance is almost identical. Under the Normal setting, all reflection models can effectively truncate reasoning paths in the general QA domain, demonstrating that our method is robust to the reasoning domains and the base reflection model. Smaller reflection models can still achieve strong results.

---

> ### Author Response · Authors · 2025-11-27
> **Official Comment to Reviewer xPcf (2/5)**
>
> **Training.** We use the same training configuration as in the main text to compare the effect of using only the length reward versus combining it with the reflection model. The prompt still uses the same setting as in the main text, i.e., “Please reason step by step, and put your final answer within \boxed{}. Question: {Question}.” We only replace the question with the concatenation of the question and choices. After training, since the dataset is large enough, we generate one path for each question and report the average results, with generation hyperparameters consistent with the main text. In addition, in SmolLM3-3B, repeated generations occasionally occur, so we add a frequency_penalty of 0.05 to alleviate this. The results demonstrate that introducing the reflection model brings stable improvements, enhancing both accuracy and efficiency.
>
> | | MMLU Acc↑ | MMLU TR↓ | MMLU Pro Acc↑ | MMLU Pro TR↓ | SuperGPQA Acc↑ | SuperGPQA TR↓ | Avg Acc↑ | Avg TR↓ |
> | ---- | -- | - | -- | --- | --- | -- | - | -- |
> | **Budget: 8k** |
> | DeepSeek-1.5B  | 40.42  | 100$_{1568}$ | 27.44  | 100$_{3632}$ | 13.52 | 100$_{4441}$  | 27.13 | 100$_{3214}$  |
> | GRPO $R_{Len}$ | 40.36  | 74.55 | 28.10  | 85.77 | 11.98 | 98.13  | 26.81 | 86.15  |
> | GRPO $R_{Len}$ + $M_{Reflect}$ | 40.94  | 73.85 | 27.88  | 89.98 | 13.56 | 93.22  | 27.46 | 85.68  |
> | SmolLM3-3B | 52.44  | 100$_{2615}$ | 34.84  | 100$_{3966}$ | 17.94 | 100$_{4369}$  | 35.07 | 100$_{3650}$  |
> | GRPO $R_{Len}$ | 52.02  | 73.61 | 34.02  | 83.59 | 16.76 | 87.48  | 34.27 | 81.56  |
> | GRPO $R_{Len}$ + $M_{Reflect}$ | 54.79  | 65.20 | 34.90  | 80.81 | 16.34 | 84.96  | 35.34 | 76.99  |
> | **Budget: 16k**  |
> | DeepSeek-1.5B  | 40.98  | 100$_{1871}$ | 30.30  | 100$_{4844}$ | 16.66 | 100$_{6337}$  | 29.31 | 100$_{4351}$  |
> | GRPO $R_{Len}$ | 40.58  | 70.82 | 30.24  | 84.56 | 15.36 | 104.92 | 28.73 | 86.77  |
> | GRPO $R_{Len}$ + $M_{Reflect}$ | 41.30  | 74.77 | 30.48  | 89.66 | 16.86 | 91.68  | 29.55 | 85.37  |
> | SmolLM3-3B | 53.54  | 100$_{2985}$ | 36.92  | 100$_{4857}$ | 19.34 | 100$_{5398}$  | 36.60 | 100$_{4413}$  |
> | GRPO $R_{Len}$ | 53.14  | 72.63 | 35.88  | 86.43 | 18.56 | 92.92  | 35.86 | 83.99  |
> | GRPO $R_{Len}$ + $M_{Reflect}$ | 55.77  | 62.04 | 36.60  | 79.16 | 18.14 | 84.94  | 36.84 | 75.38  |
>
> **Additional experiments in the math domain.** We also conduct additional experiments in the math domain. Specifically, we evaluate our approach on **DeepSeek-1.5B**, with the same training and generation config as in the main experiment. The results are as follows, showing that our approach can also generalize to smaller models in the math domain, achieving higher accuracy with comparable TR.
>
> | | GSM8k Acc↑ | GSM8k TR↓ | Math500 Acc↑ | Math500 TR↓ | GaoKao23 Acc↑ | GaoKao23 TR↓ | Amc23 Acc↑ | Amc23 TR↓ | Aime Acc↑ | Aime TR↓ | Avg Acc↑ | Avg TR↓ |
> | ---- | - | -- | --- | -- | -- | --- | - | -- | -- | - | - | -- |
> | **Budget: 8k** |
> | DeepSeek-1.5B  | 75.51 | 100$_{615}$  | 67.80 | 100$_{4137}$  | 65.97  | 100$_{3336}$ | 52.81 | 100$_{5412}$  | 18.33  | 100$_{7581}$ | 56.08 | 100$_{4216}$  |
> | GRPO $R_{Len}$ | 74.83 | 58.54  | 77.40 | 52.28  | 63.64  | 65.14 | 60.62 | 72.60  | 22.92  | 86.26 | 59.88 | 66.96  |
> | GRPO $R_{Len}$ + $M_{Reflect}$ | 72.33 | 60.81  | 80.00 | 55.14  | 64.68  | 67.51 | 65.31 | 73.50  | 24.58  | 88.83 | 61.38 | 69.16  |
> | **Budget: 16k**  |
> | DeepSeek-1.5B  | 75.59 | 100$_{622}$  | 72.00 | 100$_{5652}$  | 69.61  | 100$_{4327}$ | 57.81 | 100$_{8283}$  | 21.25  | 100$_{13358}$ | 59.25 | 100$_{6448}$  |
> | GRPO $R_{Len}$ | 74.83 | 58.84  | 78.60 | 46.16  | 65.71  | 59.88 | 63.75 | 60.69  | 24.58  | 79.45 | 61.49 | 61.00  |
> | GRPO $R_{Len}$ + $M_{Reflect}$ | 72.33 | 60.13  | 81.60 | 48.73  | 68.05  | 59.51 | 69.06 | 62.77  | 28.75  | 75.58 | 63.96 | 61.34  |
>
> Taken together with our previous results, these experiments demonstrate that our method performs well across different domains (math and general QA), different model families, and different scales (1.5B–7B). We hope these additional results alleviate the generalization concerns.
>
> [1] SmolLM3: smol, multilingual, long-context reasoner. Hugging Face 2025.
>
> [2] Qwen3 Technical Report. arXiv 2025.
>
> [3] R-Zero: Self-Evolving Reasoning LLM from Zero Data. arXiv 2025. 24 citations.
>
> [4] Measuring Massive Multitask Language Understanding. ICLR 2021.
>
> [5] MMLU-Pro: A More Robust and Challenging Multi-Task Language Understanding Benchmark. NeurIPS (Datasets & Benchmarks) 2024.
>
> [6] SuperGPQA: Scaling LLM Evaluation across 285 Graduate Disciplines. arXiv 2025.

---

> ### Author Response · Authors · 2025-11-27
> **Official Comment to Reviewer xPcf (3/5)**
>
> > W3: The method introduces extra computational overhead, requiring double rollouts and an additional reflection-model inference.
>
> Thank you for the valuable feedback. However, our method does **not** introduce double rollouts. For sequential revision, we only generate a very short suffix for labeling and the final answer, rather than a complete rollout. Although our method requires more training time, it actually consumes fewer generated tokens and has the potential to improve efficiency when deployed at scale.
>
> **Token cost analysis.** To provide a more thorough analysis of the cost for sequential revision introduced by the reflection model, we analyze token consumption. For parallel sampling, the cost of the rollout is to input the question and output a complete reasoning path. For sequential revision, we instead input the full path and output a label for each path from the reflection model, and then input the truncated path and output a very short suffix as the final answer from the policy model, and does **not** introduce double rollouts.
>
> We measure the number of tokens used to generate a single path under the two schemes during training. For parallel sampling, the average input length is 94 tokens, and the average output length is 3605 tokens on average. For sequential revision, the input is 6678 tokens and the output is 319 tokens in total. Given that our method significantly reduces the output length, this substantially reduces the cost of inference, because input tokens are typically at most 1/5 the price of output tokens. For example, for gemini-3-pro, the input price is \\$2.00 per 1M tokens (\\$0.20 for cached input), while the output price is \\$12.00 per 1M tokens; for gpt-5, the input price is \\$1.25 per 1M tokens (\\$0.125 for cached input), and the output price is \\$10.00 per 1M tokens. In particular, about half of our input tokens to the policy model are cached during parallel sampling (suffix generation by the policy model), which makes it even cheaper. Concretely, using the GPT-5 pricing as an example, parallel sampling for 1k examples costs \\$36.17, whereas sequential revision for 1k examples costs \\$7.78. This shows that, under an ideal large-scale generation scenario, the cost of our method is lower.
>
> **Latency analysis.** We also measure small-scale vLLM inference time on a **per-batch** basis, which aligns with our training setup. On a single A800 GPU, generating one batch of data with 4 paths takes 158s. Our method takes 47s to generate labels plus 2s to generate answers, for a total of 49s for sequential revision, which is less than one third of the traditional rollout. This demonstrates that the cost of sequential revision is clearly lower than that of standard parallel sampling.
>
> Therefore, although we spend slightly more total training time (110h vs 120h), we actually **reduce the generation cost** by converting expensive output tokens into cheaper input tokens. However, for online generation per batch, the three generation stages in our method must run sequentially, where each batch spends 207s (158 + 47 + 2) on inference on average. In contrast, standard generation of 4 paths takes 158s on average, and due to better parallelism, generating 8 paths takes 192s on average, which is shorter than our 207s and therefore leads to a shorter overall training time (110h vs 120h). Nevertheless, we believe that under large-scale implementations (e.g., dedicated data-generation and training servers [1-3]), the training time of our method can be lower. We hope these discussions alleviate your concerns.
>
> [1] AReaL: A Large-Scale Asynchronous Reinforcement Learning System for Language Reasoning. arXiv 2025. 37 citations.
>
> [2] LlamaRL: A Distributed Asynchronous Reinforcement Learning Framework for Efficient Large-scale LLM Training. arXiv 2025. 9 citations.
>
> [3] Echo: Decoupling Inference and Training for Large-Scale RL Alignment on Heterogeneous Swarms. arXiv 2025.

---

> ### Author Response · Authors · 2025-11-27
> **Official Comment to Reviewer xPcf (4/5)**
>
> > Q1: This work shows that the reflection model contributes to the efficiency–efficacy trade-off. Would it be possible to include a baseline that simply truncates the reasoning sequence when the first reflection token appears and then appends [object Object] to complete the response from that point?
>
> Thanks for your advice; this is an insightful suggestion. To evaluate this scheme, we add experiments on top of the main experiments. Specifically, we segment the full reasoning trace generated by DeepSeek-7B into sentences. Whenever the reflection token appears at the beginning of a sentence, we count this as one occurrence, and we truncate either at the **first** or the **fifth** such occurrence to represent different truncation strengths. We then append `</think>\n\n**Final Answers:**` and force the model to output the final answer to test the correctness of truncation, consistent with the setting in Section 3.2. The results are as follows. "First" and "Fifth" denote truncation at the first and fifth occurrences of a reflection token, respectively; "32B Revise" denotes the detection method introduced in Section 3, and "GRPO $R_{RLen+Reflect}$ $M_{Reflect}$" corresponds to our model trained in Section 4.
>
> | | GSM8K Acc↑ | GSM8K TR↓ | Math500 Acc↑ | Math500 TR↓ | Gaokao23 Acc↑ | Gaokao23 TR↓ | Amc23 Acc↑ | Amc23 TR↓ | Aime24 Acc↑ | Aime24 TR↓ | Avg Acc↑ | Avg TR↓ |
> | -- | -- | -- | -- | -- | -- | -- | -- | -- | -- | -- | -- | -- |
> | Original | 91.66 | 100$_{1344}$ | 92.00 | 100$_{3893}$ | 81.82 | 100$_{3785}$ | 88.12 | 100$_{5840}$ | 48.33 | 100$_{10460}$ | 80.39 | 100$_{5064}$ |
> | First | 69.37 | 29.84  | 49.80 | 14.98  | 43.38 | 15.67  | 29.06 | 11.92  | 2.08  | 5.87 | 38.74 | 15.66 |
> | Fifth | 89.69 | 68.01 | 70.80 | 40.41  | 60.52 | 38.57  | 55.00 | 29.76  | 16.25 | 15.50 | 58.45 | 38.45  |
> | 32B Revise | 89.46  | 62.43 | 93.20  | 71.85 | 80.52  | 70.41 | 89.06  | 79.95 | 50.83 | 93.59 | 80.61  | 75.65  |
> | GRPO $R_{RLen+Reflect}$ $M_{Reflect}$ | 89.99  | 52.08 | 91.00  | 60.78 | 82.08  | 55.85 | 89.38  | 67.76 | 51.25 | 81.09 | 80.74  | 63.51 |
>
> The results show that truncating at the **First** occurrence is too aggressive and leads to a significant drop in accuracy. This is because reflection does not only happen after the reasoning has finished; the model may also reflect during intermediate reasoning steps. When truncating at the **Fifth** occurrence, the performance on the easiest dataset (GSM8K) is close to that of **32B Revise**, but on more challenging datasets, there is still a substantial performance drop. This indicates that the optimal truncation occurrence is a difficulty-dependent hyperparameter, and choosing this parameter well can be complex. Therefore, the heuristic truncation baseline does not consistently achieve good results.
>
> Thank you again for the suggestion — we will add this baseline and the corresponding discussion to the paper, which further highlights the necessity of the reflection model.
>
> > Q2: Compared with methods using only length-based rewards, approaches involving ( M${_{reflect}}$ ) seem to yield slightly lower accuracy. Do the authors have any insights into why this happens?
>
> Thank you for your insightful observation. However, this conclusion does not hold true for all comparisons. For example, contrasting the third- and fourth-to-last rows for both the 8K and 16K budgets, introducing the reflection model **without** the reflection reward actually brings an advantage.
>
> Table 4 provides further insight into this phenomenon and suggests that it is related to the **reflection frequency**. We copy the average result as follows. Introducing the reflection model alone increases the reflection frequency, whereas adding it when a reflection reward is already present decreases the frequency. This pattern aligns with the changes in performance. Therefore, we believe there is a positive correlation between reflection frequency and model performance, but increasing the reflection frequency also increases the inference cost. Thanks again for the good question, which provides deeper insights about the experimental results.
>
> | Method | Avg Acc↑         | Avg Reflect↓ |
> |----------------------------------------------------------|------------------|--------------|
> | GRPO $R_{Len}$ | 76.88 | 255.24 |
> | GRPO $R_{Len}$ ${M}_{Reflect}$ | 79.48 | 129.56 |
> | GRPO $R_{RLen+Reflect}$ | 82.13 | 110.34 |
> | GRPO $R_{RLen+Reflect}$ ${M}_{Reflect}$ | 80.74 | 116.69 |

---

> ### Author Response · Authors · 2025-11-27
> **Official Comment to Reviewer xPcf (5/5)**
>
> > Q3: In line 319, the paper claims that GRPO with accuracy-only rewards does not further improve accuracy, which seems inconsistent with previous findings such as DeepScaleR [1]. Could the authors clarify this discrepancy or provide an explanation?
>
> Thank you for pointing this out. We believe the main reason is that the 1.5B model used in DeepScaleR has much weaker performance. In our response to Q1 and Q2, we add experiments based on the DeepSeek 1.5B model in the math domain and find that our method can also significantly improve its performance (+4.71), whereas on DeepSeek 7B, we do not observe such a large gain (+0.89 at most). This indicates that DeepSeek 1.5B has more headroom for improvement in the math domain.
>
> In addition, besides using an accuracy reward, DeepScaleR performs extensions of the context window to 16k and 24k at carefully chosen steps, which further improve results, while we only train at **8k** context. In terms of hyperparameters, they generate **eight** rollouts per question while ours generates **four**, and each of their batches contains eight times as many samples as ours. As a result, our total compute consumption is actually less than **1/8** of DeepScaleR’s. Moreover, we filter out questions that the model cannot solve within four rollouts in an 8k context, which prevents making progress on harder tasks. All of these factors limit the performance with accuracy-only rewards in our setup.
>
> It is important to emphasize that our goal is **not** to achieve the best performance under an accuracy-only reward, but rather to perform an ablation that measures how much the accuracy reward contributes under exactly the same setting. The effectiveness of our method is instead demonstrated by comparisons against the other baselines. We hope these discussions alleviate your concerns.

---

### Comment · Reviewer_1pmc · 2025-11-27
**I have receieve author's comment**

My questions are well sovled.

---

### Author Response · Authors · 2025-12-03
**Official Comment to Area Chair**

Dear Area Chair,

Thank you for overseeing the evaluation of our paper, especially given the heavy workload and tight schedule caused by the recent reassignment. Below, we outline the key points from the rebuttal to help facilitate your decision-making.

---

## 1. Summary of strengths

* Clarity

  * “Clarity is aided by step-by-step descriptions, concrete prompts, and …” (mhkM)
  * “The approach is clearly explained.” (homH)

* Motivation

  * “…targets an important and practical issue—over-reflection…” (xPcf)
  * “…an original and well-motivated idea of integrating reflection awareness …” (homH)

* Methodology

  * “... parallel sampling with sequential revision is conceptually clean, computationally practical, and …” (mhkM)
  * “… a clear revision boundary strategy and efficient implementation …” (1pmc)

* Experiments

  * “The empirical study is broad for the domain… with ablations isolating the effects of reflection modeling …” (mhkM)
  * “Experimental results are consistent and persuasive ...” (homH)

---

## 2. Key concerns and our rebuttal

### 2.1 Generalization to broader scenarios

Reviewers asked whether our method generalizes to:

* other reasoning domains (xPcf W1, mhkM W2, 1pmc W2), and
* models of different sizes, model families, and pre-trained LRMs (xPcf W2, mhkM W2, 1pmc W1).

Our rebuttal:

* We conduct new experiments on general QA reasoning, providing evidence that our reflection detection and training approach can transfer to other reasoning domains.
* We report additional results on DeepSeek-R1-Distill-Qwen-1.5B and SmolLM3-3B (non-DeepSeek, non-Qwen, pre-trained from scratch), which provide coverage of different sizes, families, and pre-trained LRMs.

### 2.2 Additional training time

Our method adds about 10% to training time (xPcf W3, mhkM W3, 1pmc W3).

Our rebuttal:

* We obtain ~3% improvement and ~4% shorter outputs, which supports the additional ~10% training time.
* Under large-scale training, our analysis suggests that it can be cheaper because it requires far fewer output tokens than a full rollout.

### 2.3 Heuristic reflective keywords

Using reflective keywords to compute overthinking signals may be too heuristic and may lead to reward hacking (homH W2, 1pmc Q2).

Our rebuttal:

* Prior work shows a strong correlation between reflective keywords and reflection behavior in pre-trained models, likely due to co-occurrence in the corpus. Increasing the frequency of reflective keywords is also an effective decoding strategy to encourage reflection, which suggests that these keywords are tightly coupled with reflection.
* To further avoid reward hacking, we carefully design the reward from three perspectives, and in our experiments, we did not observe reward hacking with this reflection reward.

### 2.4 Alternative configurations for the reflection model

* Simply truncating at the first reflection token (xPcf Q1),
* Training with only revised responses (homH Q1), and
* Handling “answer first, explain later” patterns (1pmc Q1).

Our rebuttal:

* xPcf Q1: Additional results show that even truncating at the fifth reflection token still causes a large performance drop for most benchmarks, underscoring the need for a learned reflection model.
* homH Q1: Additional results of RL training with only revised responses achieve performance similar to training with both original and revised responses, supporting the effectiveness of the revised responses.
* 1pmc Q1: For standard decoder-only LLMs, explanations appearing after the correct answer typically do not affect the prediction of that answer, and our reflection model is designed to truncate such segments.

### 2.5 Additional explanations of experimental results

* xPcf Q2: Involving ($M_{reflect}$) sometimes yields slightly lower accuracy. However, this effect is not uniform across settings; our analysis suggests that the changes in performance are related to the reflection frequency.

* xPcf Q3: Our accuracy-only rewards do not significantly improve accuracy. Compared with DeepScaleR, their system uses a 1.5B base model (more headroom for improvement, as suggested by additional results on DeepSeek 1.5B), substantially more compute, and context window extensions. Besides, our accuracy-only reward experiments are intended as ablations, not our primary route to performance gains.

* mhkM W4: Budget-equalized comparisons are difficult to control across baselines. In our experiments, our methods either provide both higher efficiency and better performance, or yield substantial gains (8–16 points under 16k budget) compared to baselines. We also analyze budget-equalized checkpoints for the length-reward-only method.

* homH W1: Adding the gold answer does not lead to higher accuracy for revision. It typically uses fewer tokens, thus less information under truncation settings. Moreover, the gold answer itself appears to add limited extra information, as in our analysis and case studies.

---

Best regards,

*Authors of Paper 15536*

---

### Meta-Review · Area_Chair_9aHR · 2026-01-08

**Summary:**

This paper proposes REA-RL, a reflection-aware online reinforcement learning framework aimed at improving the efficiency of reasoning.
Most reviewers responded positively, although one raised concerns regarding the experimental evaluation. The area chair finds the work interesting and has therefore decided to recommend its acceptance.

**Reviewer Concerns:**

The experiments need to be more extensive. The rebuttal addressed this concern.

**Reviewer Scores:**

The rebuttal is solid.

---

### Decision · Program_Chairs · 2026-01-26

Accept (Poster)